



Atmospheric
Measurement
Techniques

# An overview of and issues with sky radiometer technology and SKYNET

**Teruyuki Nakajima**[1], **Monica Campanelli**[2], **Huizheng Che**[3], **Victor Estellés**[2,4], **Hitoshi Irie**[5], **Sang-Woo Kim**[6], **Jhoon Kim**[7], **Dong Liu**[8], **Tomoaki Nishizawa**[9], **Govindan Pandithurai**[10], **Vijay Kumar Soni**[11], **Boossarasiri Thana**[12], **Nas-Urt Tugjsurn**[13], **Kazuma Aoki**[14], **Sujung Go**[7,15], **Makiko Hashimoto**[1], **Akiko Higurashi**[9], **Stelios Kazadzis**[16], **Pradeep Khatri**[17], **Natalia Kouremeti**[16], **Rei Kudo**[18], **Franco Marenco**[19], **Masahiro Momoi**[5,20], **Shantikumar S. Ningombam**[21], **Claire L. Ryder**[22], **Akihiro Uchiyama**[9], and **Akihiro Yamazaki**[18]

[1]Satellite Observation Center, National Institute for Environmental Studies, 16-2 Onogawa, Tsukuba 305-8506, Japan
[2]Consiglio Nazionale delle Ricerche, Istituto Scienze dell'Atmosfera e del Clima,
via Fosso del Cavaliere 100, 00133, Rome, Italy
[3]Centre for Atmosphere Watch And Services, CMA Chinese Academy of Meteorological Sciences,
46 Zhong-Guan-Cun S. Ave., Beijing 100081, China
[4]Dept. Física de la Terra i Termodinamica, Universitat de Vale'ncia, Burjassot, Valencia, Spain
[5]Center for Environmental Remote Sensing, Chiba University, Chiba 263-8522, Japan
[6]School of Earth and Environmental Sciences, Seoul National University, Seoul 08826, Republic of Korea
[7]Dept. of Atmospheric Sciences, Yonsei University, Seoul 03722, Republic of Korea
[8]Center for Atmospheric Optics, Anhui Institute of Optics and Fine Mechanics, Hefei Institutes of Physical Science,
Chinese Academy of Sciences, Hefei, Anhui 230031, China
[9]National Institute for Environmental Studies, 16-2 Onogawa, Tsukuba, Ibaraki 305-8506, Japan
[10]Indian Institute of Tropical Meteorology, Ministry of Earth Sciences, Pune 411 008, India
[11]Environment Monitoring & Research Centre, India Meteorological Department, Ministry of Earth Sciences,
Mausam Bhawan, Lodi Road, New Delhi 110 003, India
[12]Thailand Global Warming Academy, Napamitr Foundation, 234/88 Asoke-Din Daeng Road, Bang Kapi Sub-district,
Huai Khwang District, Bangkok 10310, Thailand
[13]Physics department, Mongolian University of Science and Technology, 216046, Ulaanbaatar, Mongolia
[14]Graduate School of Science and Engineering (Science), University of Toyama, 3190 Gofuku, Toyama 930-8555, Japan
[15]Joint Center for Earth Systems Technology (JCET), University of Maryland Baltimore County (UMBC),
Baltimore, MD 21228, USA
[16]Physikalisch-Meteorologisches Observatorium Davos, World Radiation Center,
Dorfstrasse 33, 7260 Davos, Switzerland
[17]Center for Atmospheric and Oceanic Studies, Graduate School of Science, Tohoku University, Sendai, Japan `CE1`
[18]Meteorological Research Institute, Meteorological Agency, Nagamine, Tsukuba, Ibaraki 305-0052, Japan
[19]Space Applications and Nowcasting, Met Office, UK `TS1`
[20]Graduate School of Science, Tokyo University of Science, Tokyo 162-8601, Japan
[21]Indian Institute of Astrophysics, 2nd Block Koramangala, Bangalore 560 034, India
[22]Department of Meteorology, University of Reading, Reading, RG6 6BB, UK

**Correspondence:** Teruyuki Nakajima (terry-nkj@nifty.com) `TS2`

Received: 2 March 2020 – Discussion started: 16 March 2020
Revised: 22 June 2020 – Accepted: 29 June 2020 – Published:

**Published by Copernicus Publications on behalf of the European Geosciences Union.**

**Abstract.** This paper is an overview of the progress in sky radiometer technology and the development of the network called SKYNET. It is found that the technology has produced useful on-site calibration methods, retrieval algorithms, and data analyses from sky radiometer observations of aerosol, cloud, water vapor, and ozone.

A formula was proposed for estimating the accuracy of the sky radiometer calibration constant $F_0$ using the improved Langley (IL) method, which was found to be a good approximation to observed monthly mean uncertainty in $F_0$, around 0.5 % to 2.4 % at the Tokyo and Rome sites and smaller values of around 0.3 % to 0.5 % at the mountain sites at the IOA CE2 and Davos. A new cross IL (XIL) method was also developed to correct an underestimation by the IL method in cases with large aerosol retrieval errors.

The root-mean-square difference (RMSD) in aerosol optical thickness (AOT) comparisons with other networks took values of less than 0.02 for $\lambda \geq 500$ nm and a larger value of about 0.03 for shorter wavelengths in city areas and smaller values of less than 0.01 in mountain comparisons. Accuracies of single-scattering albedo (SSA) and size distribution retrievals are affected by the propagation of errors in measurement, calibrations for direct solar and diffuse sky radiation, ground albedo, cloud screening, and the version of the analysis software called the Skyrad pack. SSA values from SKYNET were up to 0.07 larger than those from AERONET, and the major error sources were identified as an underestimation of solid viewing angle (SVA) and cloud contamination. Correction of these known error factors reduced the SSA difference to less than 0.03.

Retrievals of other atmospheric constituents by the sky radiometer were also reviewed. Retrieval accuracies were found to be about 0.2 cm for precipitable water vapor amount and 13 DU (Dobson Unit) for column ozone amount. Retrieved cloud optical properties still showed large deviations from validation data, suggesting a need to study the causes of the differences.

It is important that these recent studies on improvements presented in the present paper are introduced into the existing operational systems and future systems of the International SKYNET Data Center. CE3

## 1 Introduction

A sun–sky radiometer is a narrow-band filter photometer able to perform measurements of direct solar and diffuse sky radiation at selected wavelengths and at several scattering angles. Observed data have large information content for aerosol, cloud, and gaseous constituents but are difficult to retrieve because of the need for full radiative transfer computation to quantify single- and multiple-scattered radiation.

The origin of the idea of the technology dates back to the beginning of the last century (Shaw, 2006). Long-term direct solar and diffuse sky measurements were carried out during 1923–1957 by the Smithsonian Astronomical Observatory by monitoring the solar constant with a pyrheliometer at Montezuma (Chile) and Table Mountain (California) (Abbot, 1911; Ångström, 1961, 1974; Roosen et al., 1973; Hoyt, 1979a, b). Diffuse sky irradiance in the circumsolar or solar aureole region was measured by the pyranometer to correct for the atmospheric effects in the measured solar constant (Abbot and Aldrich, 1916). This method was also used by Kalitin TS3 (1930), Fesenkov (1933), and Pyaskovskaya-Fesenkova (1957) (Terez and Terez, 2003). By the 1970s, spectral measurements of the direct solar radiation became popular for air pollution monitoring via the introduction of a low-cost compact narrow-band radiometer called a sun photometer, with a silicon photodiode and cutoff or interference optical filters (Volz, 1959, 1974). In parallel, pioneering measurements of spectral diffuse sky radiance started from the ground and aircraft (Bullrich, 1964; Bullrich et al., 1967, 1968; Murai, 1967; Eiden, 1968; Green et al., 1971; Gorodetskiy et al., 1976; Twitty et al., 1976). They were attracted CE4 by the characteristic radiance distributions, including bright circumsolar region and neutral points of the degree of polarization in the sky dome. Theoretical and inversion schemes for the involved ill-conditional problems were studied for data analysis (Deirmendjian, 1957, 1959; Phillips, 1962; Twomey 1963; de Bary, 1964; Turchin and Nozik, 1969; Yamamoto and Tanaka, 1969; Dave, 1971; Shifrin et al., 1972; Shifrin and Gashko, 1974).

By the 1980s, analyses of combined sun and sky radiation data became comprehensive (e.g., O'Neill and Miller, 1984a, b; Tanaka et al., 1986; Tanré et al., 1988) after full yet fast radiative transfer computation became possible, allowing quantification of the multiple-scattering component of sky radiance and retrieval of the column-averaged size distribution and the complex refractive index of polydispersed aerosol (Twitty, 1975; Weinman et al., 1975; Box and Deepak, 1978, 1979; Nakajima et al., 1983; O'Neill and Miller, 1984b; Tanré et al., 1988; Tonna et al., 1995; Dubovik and King, 2000; Dubovik et al., 2000, 2002). Networks of radiometers have been developed to utilize sun and sky measurement data for various applications, such as satellite remote sensing validation, air pollution monitoring, and the study of the climate effects of atmospheric constituents, as overviewed by Holben et al. (2001). The largest network is NASA AERONET (Holben et al., 1998) developed in the early 1990s and currently with more than 500 sun–sky radiometers. Later, in the 2000s SKYNET was formed with sky radiometers (Nakajima et al., 2007). Compared to the AERONET technology, SKYNET has several differences in measurement and analysis methods.

SKYNET is for research purposes without a centralized data analysis system and its information is scattered in independent papers and documents, which makes SKYNET difficult to understand for the science community. As a result, this paper intends to put forward an overview of the key findings

of and issues with SKYNET to provide better information for the community.

## 2 Sun and sky measurements from the sky radiometer

SKYNET is a research group of users of sky radiometers that was initiated around the time of the East Asian Regional Experiment (EAREX) 2005 (Nakajima et al., 2007), one of the regional experiments under the UNEP Atmospheric Brown Cloud (ABC) project (Ramanathan et al., 2007). A number of sky radiometers were deployed in the East Asian region for measuring aerosol optical properties in order to estimate the aerosol impact on the Earth's radiation budget (Takamura et al., 2004; Khatri et al., 2010). Since then, users of sky radiometers have kept growing globally and the number of sky radiometers now exceeds 100 units. Table 1 and Fig. 1 show the sky radiometer sites as recognized by the International SKYNET Committee (ISC). Users established regional subnetwork groups in China, Europe, India, Japan, South Korea, Mongolia, and Southeast Asia for data analysis and formed the ISC to discuss international collaboration issues (Fig. 2). Historically, two major groups were grown for regional data collection and analysis: the SR-Center for Environmental Remote Sensing (SR-CEReS) of Chiba University (Takamura et al., 2004, 2009, 2013) and the European SKYNET Radiometers network (ESR) (Campanelli et al., 2004, 2007, 2012). Analysis systems were developed by the sub-networks independently, and thus analysis methods and data archive systems have not been unified. Table 2 lists the archived geophysical parameters, versions of the retrieval software called the Skyrad pack, and the data availability in the known data archives. Table 2 indicates that the features of the archives are different from each other and difficult to use for the science community.

In 2017, SKYNET became a contributing network of the WMO Global Atmospheric Watch (GAW) (https:// community.wmo.int/activity-areas/gaw TS5). In this expanding situation of SKYNET having more burden and responsibility, the ISC decided to establish the International SKYNET Data Center (ISDC) at the National Institute for Environmental Studies (NIES) in Japan to start a shared data collection and analysis based on the MOU CE7 between users and the ISDC. Among the sites in Table 1, the ISDC started receiving data from 25 sites across the world. The ISDC is going to provide standard products from the SKYNET network, whereas the regional sub-networks will develop new research products and test new methodologies.

The main instrument of SKYNET is the sky radiometer manufactured by PREDE Co., Ltd. Several versions of the radiometer have been made at user requests. POM-01 is the standard version, with seven wavelengths of $\lambda = 315, 400, 500, 675, 870, 940,$ and $1020$ nm, and POM-02 is an extended version, with UV wavelengths of 340 and 380 nm and shortwave infrared wavelengths of 1600 and 2200 nm.

Channels of 315 and 940 nm are installed for ozone and water vapor amount retrievals. Full widths at half maximum of band-pass filters are 3 nm or less for channels shorter than 380 nm, 10 nm between 400 and 940 nm, and 20 nm for longer wavelengths. There is a modified version of POM-02 for lunar photometry (Uchiyama et al., 2019). Shipborne versions have been also built (Kobayashi and Shiobara, 2015).

Sky radiometer readings of direct solar and diffuse sky measurements, $V_d$ and $V_s$, are related to the direct solar irradiance $F_d$ and sky radiance $\overline{L}_s$ at the mean Earth orbit as follows:

$$F_d = C_R R_{es}^2 V_d, \quad \overline{L}_s = \frac{F_s}{\Delta\Omega} = C_R \frac{R_{es}^2 V_s}{\Delta\Omega}, \tag{1}$$

where $C_R$ is the radiometric sensitivity or calibration coefficient of the radiometer to translate the radiometer reading to irradiance unit, e.g., $W\,m^{-2}\,nm^{-1}$; $\Delta\Omega$ is the solid viewing angle (SVA) of the radiometer; and $R_{es}$ is the sun–Earth distance in astronomical units. SKYNET remote sensing uses the beam transmittance $T_d$ of the atmosphere and relative sky radiance $R$ (Nakajima et al., 1986)

$$T_d \equiv \frac{F_d}{F_0} = \exp(-m_0 \tau), \tag{2a}$$

$$\tau = \tau_a + \tau_m, \quad \omega\tau = \omega_a \tau_a + \omega_m \tau_m, \tag{2b}$$

$$R(\theta, \phi; \theta_0, \phi_0) \equiv \frac{\overline{L}_s(\theta, \phi; \theta_0, \phi_0)/m}{F_d} = \frac{1}{m\Delta\Omega}\frac{V_s}{V_d}, \tag{2c}$$

where $\tau$ is the optical thickness (OT) of the atmosphere consisting of molecular optical thickness $\tau_m$, single-scattering albedo (SSA) $\omega_m$, aerosol optical thickness (AOT) $\tau_a$, and SSA $\omega_a$ in the clear sky condition; $F_0$ is the extraterrestrial solar irradiance (ESI); $(\theta_0, \varphi_0)$ and $(\theta, \varphi)$ are zenith and azimuthal angles of the sun and the line of sight of the sky radiometer, respectively; $m_0$ and $m$ are optical air masses for solar insolation and the line of sight of the radiometer, which are approximated as $1/\cos(\theta_0)$ and $1/\cos(\theta)$ for plane-parallel geometry of the atmosphere. SKYNET adopts on-site calibration routines to determine the two radiometric constants, $F_0$ and $\Delta\Omega$, using the improved Langley plot method (hereafter, IL or ILP CE8) and the disk scan method (Nakajima et al., 1996; Boi et al., 1999; Uchiyama et al., 2018a, b TS6), as discussed in Sect. 3 and 4. Under the condition that $C_R$ and $F_0$ do not change between the time of measurement and time of $F_0$ determination, $T_d$ and $R$ do not depend on the calibration coefficient $C_R$, and thus we can select the radiometer reading for $F_0$, i.e., $C_R = 1$, without the absolute radiometric calibration. Under this assumption, $F_0$ in the radiometer reading is sometimes called a calibration constant. In order to meet this condition, the on-site calibration is required to be performed as frequently as possible to monitor change of $C_R$ due to machine condition change and $F_0$ change due to solar luminosity change.

Standard measurement protocols of SKYNET are as follows. Direct solar irradiance is measured every 1 min.

**Table 1.** Sites recognized by the International SKYNET Committee. CE5

| Name | Owner | Country | Location (lat) | Location (long) | Location (m a.m.s.l.) | Sub-net analyzers | ISDC |
|---|---|---|---|---|---|---|---|
| Halley | British Antarctic Survey | Antarctica | 75.350° S | 26.340° W | 30 m | ESR | |
| Rothera | British Antarctic Survey | Antarctica | 67.340° S | 68.080° W | 0 m | ESR | |
| Showa | NIPR | Antarctica | 69.006° S | 39.590° E | 30 m | – | x |
| Chajnantor, Atacama | Universidad de Santiago de Chile | Chile | 33.451° S | 70.686° W | 5100 m | ESR | |
| Beijing/CAMS | CMA | China | 39.933° N | 116.317° E | 106 m | CAMS | |
| Bejing/IAP | IAP-CAS | China | 39.977° N | 116.381° E | 92 m | CAS | |
| Dunhuang | IAP-CAS | China | 40.146° N | 90.799° E | 1120 m | CAS, Chiba-U | |
| Hefei | AIOFM-CAS | China | 31.897° N | 117.173° E | 30 m | CAS, U-Toyama | |
| Lanzhou | Lanzhou-U | China | 35.570° N | 104.133° E | 1965 m | Lanzhou-U | |
| Qionghai | IAP-CAS | China | 19.230° N | 110.46° E | 24 m | CAS | |
| Xi'an | XAUT | China | 34.25° N | 108.983° E | 396.9 m | XAUT | |
| Orleans | NIES | France | 47.965° N | 2.113° E | 131 m | Chiba-U | x |
| Lindenberg | Meteorologisches Oberva-torium Lindenberg (Mark) | Germany | 52.209° N | 14.121° E | 120 m | ESR | |
| Amaravati | IMD | India | 16.573° N | 80.358° E | 343 m | IMD | |
| Aurangabad | IMD | India | 19.876° N | 75.343° E | 568 m | IMD | |
| Gangtok | IMD | India | 27.339° N | 88.607° E | 1650 m | IMD | |
| Guwahati | IMD | India | 26.100° N | 91.580° E | 54 m | IMD | |
| Hanle | Indian Institute of Astrophysics | India | 32.779° N | 78.964° E | 4500 m | IIAP, Chiba-U | x |
| Hyderabad | National Remote Sensing Agency, India | India | 17.469° N | 78.486° E | 811 m | IMD, U-Toyama | |
| Jaipur | IMD | India | 27.175° N | 75.955° E | 431 m | IMD | |
| Jodhpur | IMD | India | 26.300° N | 73.020° E | 224 m | IMD | |
| Kolkata | IMD | India | 22.650° N | 88.450° E | 88 m | IMD | |
| Merak | Indian Institute of Astrophysics | India | 33.480° N | 78.360° E | 4258 m | IIAP, Chiba-U | x |
| Minicoy | IMD | India | 8.274° N | 73.050° E | 2 m | IMD | |
| Nagpur | IMD | India | 21.100° N | 79.050° E | 310 m | IMD | |
| New Delhi/IITM | Indian Institute of Tropical Meteorology | India | 28.629° N | 77.174° E | 240 m | IMD, Chiba-U | |
| New Delhi/IMD | IMD | India | 28.580° N | 77.210° E | 216 m | IMD, U-Toyama | |
| New Delhi/NPL | National Physical Laboratory, India | India | 28.637° N | 77.174° E | 223 m | IMD, U-Toyama | |
| Port Blair | IMD | India | 11.670° N | 92.720° E | 79 m | IMD | |
| Puducherry | IMD | India | 11.942° N | 79.808° E | 3 m | IMD | |
| Pune/IITM | Indian Institute of Tropical Meteorology | India | 18.537° N | 73.805° E | 559 m | IMD, Chiba-U | |
| Pune/IMD | IMD | India | 18.530° N | 73.850° E | 559 m | IMD | |
| Raipur | IMD | India | 21.251° N | 81.630° E | 298 m | IMD | |
| Ranichauri | IMD | India | 30.250° N | 78.080° E | 1800 m | IMD | |
| Rohtak | IMD | India | 28.830° N | 76.580° E | 214 m | IMD | |
| Sagar | IMD | India | 23.839° N | 78.738° E | 427 m | IMD | |
| Trivandrum | IMD | India | 08.480° N | 76.950° E | 60 m | IMD | |
| Varanasi | IMD | India | 25.300° N | 83.020° E | 90 m | IMD | |
| Visakhapatnam | IMD | India | 17.720° N | 83.230° E | 18 m | IMD | |
| Aosta | ARPA-VDA | Italy | 45.742° N | 7.357° E | 570 m | ESR | |
| Bologna | CNR-ISAC | Italy | 44.650° N | 11.650° E | 8 m | Chiba-U | |
| Bologna | CNR-ISAC | Italy | 44.520° N | 11.340° E | 60 m | Chiba-U | |
| Messina | Italian Air force | Italy | 38.200° N | 15.500° E | 0 m | ESR | |
| Monte Cimone | Italian Air force | Italy | 44.190° N | 10.700° E | 2165 m | ESR | |
| Novara | Italian Air force | Italy | 45.530° N | 8.670° E | 169 m | ESR | |
| Paganella | Italian Air force | Italy | 46.110° N | 11.040° E | 2129 m | ESR | |
| Rome | CNR-ISAC | Italy | 41.905° N | 12.548° E | 70.0 m | ESR | |
| Sigonella | Italian Air force | Italy | 37.405 ° N | 14.919 ° E | 30 m | ESR | |
| Vigna di Valle | Italian Air force | Italy | 42.080° N | 12.210° E | 270 m | ESR | |

| Name | Owner | Country | Location (lat) | Location (long) | Location (m a.m.s.l.) | Sub-net analyzers | ISDC |
|---|---|---|---|---|---|---|---|
| Abashiri | U-Toyama | Japan | 44.018° N | 144.280° E | 45 m | U-Toyama | |
| Chiba | Chiba-U | Japan | 35.625° N | 140.104° E | 21 m | Chiba-U, U-Toyama | x |
| Etchujima | Tokyo Univ. Marine Sci. | Japan | 35.664° N | 139.796° E | 35.0 m | Chiba-U, U-Toyama | x |
| Fuji Hokuroku | AIST | Japan | 35.433° N | 138.750° E | 1150 m | Chiba-U, U-Toyama | x |
| Fukue | Chiba-U | Japan | 32.752° N | 128.682° E | 80 m | Chiba-U, U-Toyama | x |
| Fukuoka | Kyushu-U | Japan | 33.524° N | 130.475° E | 28 m | Chiba-U, U-Toyama | x |
| Fukuoka | MRI | Japan | 33.552° N | 130.365° E | 31 m | MRI | |
| Fussa | U-Toyama | Japan | 35.751° N | 139.323° E | 141 m | U-Toyama | |
| Hedo | Chiba-U | Japan | 26.867° N | 128.248° E | 65 m | Chiba-U, U-Toyama | x |
| Ishigaki | JMA | Japan | 24.337° N | 124.164° E | 6 m | JMA | x |
| Itabashi | Tokyo Kasei Univ. | Japan | 35.775° N | 139.721° E | 70 m | U-Toyama | |
| Jodo | U-Toyama | Japan | 36.566° N | 137.606° E | 2839 m | U-Toyama | |
| Kamiyukawa | U-Toyama | Japan | 34.062° N | 135.516° E | 535 m | U-Toyama | |
| Kanazawa | Kanazawa Institute of Technology | Japan | 36.533° N | 136.629° E | 26 m | U-Toyama | |
| Kofu | Chiba-U | Japan | 35.650° N | 138.567° E | 300 m | Chiba-U, U-Toyama | x |
| Minamitorishima | MRI | Japan | 24.300° N | 153.970° E | 7 m | MRI | |
| Minamitorishima/JMA | JMA | Japan | 24.288° N | 153.983° E | 7 m | JMA | x |
| Miyakojima | MRI | Japan | 24.737° N | 125.327° E | 50 m | MRI, Chiba-U | |
| Moshiri | NIES | Japan | 44.366° N | 142.260° E | 288 m | Chiba-U, U-Toyama | x |
| Nagasaki | Nagasaki-U | Japan | 32.786° N | 129.865° E | 35 m | Chiba-U, U-Toyama | |
| Okayama | AIST | Japan | 34.664° N | 133.931° E | 13 m | U-Toyama | |
| Osaka | Kinki-U | Japan | 34.642° N | 135.587° E | 19 m | Chiba-U, U-Toyama | |
| Saga | Saga-U, NIES | Japan | 33.233° N | 130.283° E | 8 m | Chiba-U | x |
| Sapporo/ILTS | U-Toyama | Japan | 43.084° N | 141.339° E | 30 m | U-Toyama | |
| Sapporo/ILTS,MRI | U-Toyama | Japan | 43.084° N | 141.339° E | 30 m | U-Toyama | |
| Sapporo/JMA | JMA | Japan | 43.060° N | 141.329° E | 17 m | JMA | x |
| Sendai | Tohoku-U | Japan | 38.260° N | 140.840° E | 153 m | Chiba-U, U-Toyama | x |
| Shigaraki | U-Toyama | Japan | 34.854° N | 136.105° E | 295 m | U-Toyama | |
| Suzu, Ishikawa | U-Toyama | Japan | 37.451° N | 137.359° E | 15 m | U-Toyama | |
| Takayama | Gifu-U | Japan | 36.145° N | 137.423° E | 1420 m | Chiba-U, U-Toyama | x |
| Takikawa | U-Toyama | Japan | 43.547° N | 141.897° E | 40 m | U-Toyama | |
| Toyama | U-Toyama | Japan | 36.700° N | 137.187° E | 30 m | U-Toyama | |
| Tsukuba | Tsukuba-U | Japan | 36.114° N | 140.096° E | 27 m | Chiba-U, U-Toyama | x |
| Tsukuba/MRI | MRI | Japan | 36.056° N | 140.125° E | 30 m | MRI | |
| Kagurazaka | Tokyo Univ. of Science | Japan | 35.699° N | 139.741° E | 70 m | – | |
| Mandargovi | Chiba-U | Mongolia | 45.743° N | 106.264° E | 1393 m | Chiba-U | x |
| Ulaanbaatar | MUST, Chiba-U | Mongolia | 47.886° N | 106.906° E | 1350 m | Chiba-U | x |
| Lauder | NIWA, NIES | New Zealand | 45.038° S | 169.681° E | 370 m | Chiba-U | x |
| Ny-Alesund | NIPR | Norway | 78.930° N | 11.861° E | 50 m | U-Toyama | x |
| Belsk | Polish Academy of Science | Poland | 51.837° N | 20.792° E | 190 m | ESR, U-Toyama | |
| Anmyon | SNU | Republic of Korea | 36.517° N | 126.317° E | 45 m | SNU, U-Toyama | |
| Kongju | Kongju National Univ. | Republic of Korea | 36.280° N | 127.080° E | 70 m | SNU, U-Toyama | |
| Seoul | SNU | Republic of Korea | 37.460° N | 126.950° E | 150 m | SNU, Chiba-U, U-Toyama | x |
| Yongin | Hankuk University of Foreign Studies | Republic of Korea | 37.336° N | 127.268° E | 167 m | SNU, U-Toyama | x |
| Yonsei | Yonsei-U | Republic of Korea | 37.570° N | 126.980° E | 60 m | SNU | x |
| Barcelona | Universitat de Barcelona | Spain | 41.385° N | 2.118° E | 97 m | ESR | |
| Valencia-Burjassot | Universitat de Valencia | Spain | 39.507° N | 0.420° W | 60 m | ESR, Chiba-U | |
| Bangkok | TMD | Thailand | 13.667° N | 100.605° E | 60 m | Chiba-U | |
| Phimai | Chiba-U | Thailand | 15.184° N | 102.565° E | 212 m | Chiba-U, U-Toyama | x |
| Sri-samrong | Chiba-U | Thailand | 17.157° N | 99.867° E | 50 m | Chiba-U, U-Toyama | |
| Cambridge | British Antarctic Survey | United Kingdom | 52.215° N | 0.080° E | 30 m | ESR | |
| Cardington | Met-Office | United Kingdom | 52.100° N | 0.421° W | 30 m | ESR | |
| London | University College London-UAO | United Kingdom | 51.524° N | 0.131° W | 45 m | ESR | |
| Plymouth | Plymouth Marine Lab. | United Kingdom | 50.366° N | 4.148° W | 0 m | ESR | |
| Aurora, Colorado | AIST | USA | 39.400° N | 104.500° W | 1674 m | ESR | |
| Golden | National Renewable Energy Laboratory | USA | 39.740° N | 105.180° W | 1829 m | ESR | |

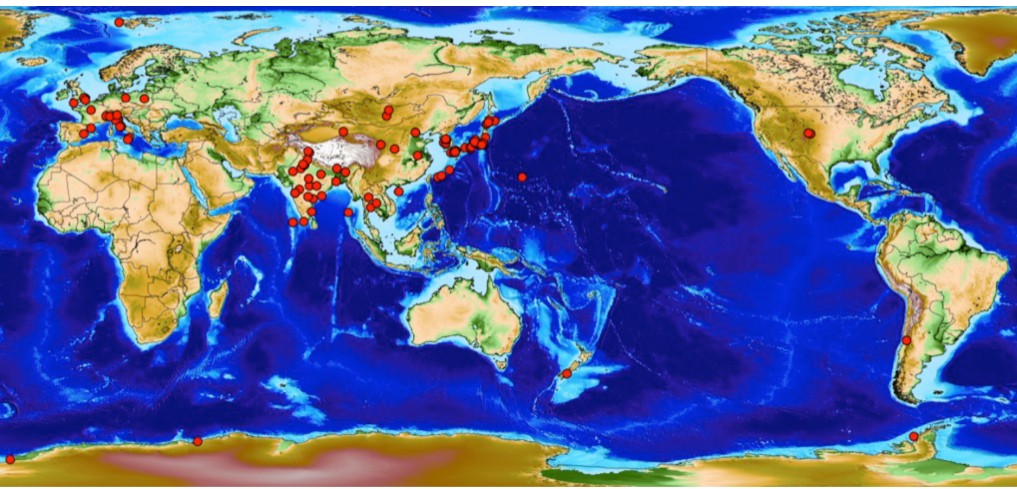

**Figure 1.** A map of the sky radiometer sites.

**Table 2.** Geophysical parameter products, versions of Skyrad pack, and availability of the known data archives. CE6 TS4

| ESR:<br>http://www.euroskyrad.net | L2 products: AOT, AE, SDF, SSA, CRI, phase function, asymmetry factor, lidar ratio, linear depolarization ratio<br>Analysis software: Skyrad pack v4.2, MRI v2<br>Data availability: L2 data are open via the web system |
|---|---|
| SR-CEReS:<br>http://atmos3.cr.chiba-u.jp/skynet/ | L2 products: AOT, AE, SDF, SSA, CRI<br>Analysis software: Skyrad pack v5<br>Data availability: L2 data are open via the web system |
| Toyama U:<br>http://skyrad.sci.u-toyama.ac.jp/ | L2 products: AOT, AE, SDF, SSA, CRI<br>Analysis software: Skyrad pack v4.2, 5<br>Data availability: L2 data are open via individual request |
| MRI | L2 products: AOT, AE, SDF, SSA, CRI, phase function, asymmetry factor, lidar ratio, linear depolarization ratio<br>Analysis software: Skyrad pack MRI v1 and v2<br>Data availability: L2 data are available from PIs upon request |
| CAMS-SKYNET | Operational L2 products: AOT, AE, SDF, SSA, CRI<br>Analysis software: Skyrad pack v4.2 and 5<br>Data availability: L2 data are used by CMA, data are available from PIs upon request |
| IMD | Operational L2 products: AOT, AE, SDF, SSA<br>Analysis software: Skyrad pack v4.2<br>Data availability: L2 data are used by IMD, no open web system |

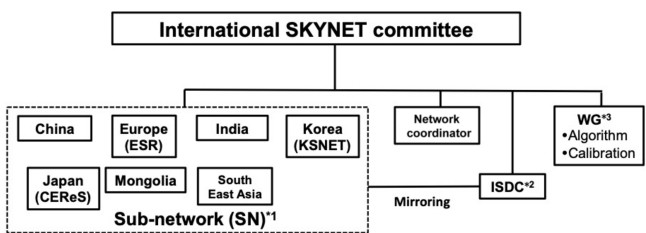

**Figure 2.** The structure of the International SKYNET Committee.

Diffuse sky radiance is measured by full almucantar scan at scattering angles from the set of $\Theta = \{2°(1°)5°, 7°, 10°(5°)30°(10°)160°\}$ for $\theta_0 \leq 78°$, whereas the forward almucantar scan is made in $\Theta \leq 30°$ for obtaining quick scan data for ILP and/or in a condition of rapid air mass change for $\theta_0 > 78°$. AERONET adopts a two-side scan of the sun for a symmetry check and spatial averaging of sky radiances to minimize the inhomogeneous effects. On the other hand, SKYNET basically uses a one-side almucantar scan of the sun to save on observation time. At some

sites, however, the almucantar scan is made on either side of the sun alternatively, and the retrieval is made for each side separately to evaluate inhomogeneous aerosol distribution in space and time. The sky radiometer has several angle scan modes, i.e., almucantar scan, principal plane scan, cloud scan, and solar disk scan. There are two temporal sampling modes of a regular time interval of 10 min (mode 1) and a regular solar air mass interval of 0.25 (mode 2). Most of the sites in Table 1 adopt the mode 1 measurement with a one-side almucantar scan. The disk scan mode is scheduled once a week at 10:00 LT, though the scan time can be changed according to the user's plan. A cloud scan mode at nadir is taken every 10 min at POM-02 sites and some POM-01 sites.

Once the radiometric constants are determined, the direct solar irradiance $F$ and relative sky radiance $R$ are used for the level 2 (L2) analysis, i.e., retrievals of the geophysical parameters of aerosol, cloud, water vapor, and ozone, as discussed later in Sect. 5. The flow of sky radiometer measurements and data analysis are schematically depicted in Fig. 3. As overviewed in the following sections, $F_0$ and SVA are obtained on-site through various Langley plot methods and the solar disk scan method using data from direct solar and forward scan measurements. Cloud screening is also performed differently by different sub-networks. The ESR performs a cloud screening for a direct solar measurement at 1 min frequency using a procedure based on the methodology developed by Smirnov et al. (2000; Estellés et al., 2012a; Song et al., 2014). Cloud screening for sky measurements uses the downward shortwave radiative flux measured by a co-located pyranometer (Khatri and Takamura, 2009), otherwise they do not perform cloud screening for sky data. CEReS conducts the cloud screening with the method of Khatri and Takamura (2009) but without using global irradiance data from a pyranometer (Irie et al., 2019). It corresponds to the combination of a spectral variability test (Kaufman et al., 2006) and statistical analysis test of Smirnov et al. (2000) including checking the amount of data, the diurnal stability check, smoothness criteria, and three standard deviation criteria but without a triplet stability criteria test. We do not use several QC CE9 tests, such as the angular steepness of the solar aureole, for a stricter cloud filter as is done for AERONET (Giles et al., 2019).

To obtain data for L2 data analysis for retrieval of geophysical parameters for atmospheric constituents, an analysis software called the Skyrad pack has been developed (Nakajima et al., 1996; Hashimoto et al., 2012) and is publicly available on the OpenCLASTR shareware site (http: //157.82.240.167/~clastr/data_policy.html TS7) for use by the research community. Various L2 products are retrieved by the Skyrad pack, such as spectra of AOT, its Ångström exponent (AE), size distribution function (SDF), SSA, complex refractive index (CRI) , asphericity, cloud optical thickness (COT), cloud effective particle radius (CER), water and ice phase from data in the non-gas-absorbing channels, precipitable water vapor (PWV), and column ozone amount ($O_3$) from the

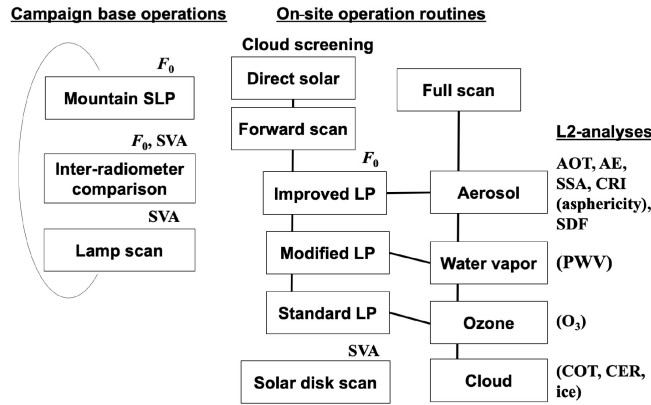

**Figure 3.** A flowchart of the SKYNET analysis. Quantities in parentheses are research products.

gas-absorbing channels, as explained in the following sections. Common operational products of the sub-networks are AOT, AE, SDF, SSA, and CRI-assuming Mie particles. Other products have been retrieved by research studies. The current operating versions are version 4.2 and 5, and a version from the Meteorological Research Institute of Japan Meteorological Agency (MRI version) developed by Kobayashi et al. (2006, 2010).

## 3   Radiometric calibration of the direct solar irradiance measurements

In the case of non-gas-absorption channels, the standard Langley plot method (SL or SL plot method) can be used to obtain $F_0$ by plotting the logarithm of the Lambert–Beer's law Eq. (2a) versus $m_0$,

$$\ln(F_d) = \ln(F_0) - m_0\tau, \tag{3}$$

to extrapolate the linear regression line to $m_0 = 0$. It is known, however, that an air mass dependence or a quadratic time dependence of AOT introduces a serious error in the SL, as claimed by Shaw (1976). Correction methods to this problem were proposed by O'Neill and Miller (1984a, b) and Tanaka et al. (1986) with the use of a time dependence of the circumsolar radiance of which the major part is approximated by the single-scattered radiance proportional to the OT along the solar almucantar circle ($\theta = \theta_0$), given as follows:

$$R(\theta, \phi; \theta_0, \phi_0) = \omega\tau P(\Theta) + R_{\text{mult}}(\theta, \phi; \theta_0, \phi_0), \tag{4}$$

where $P$ is the normalized scattering phase function at the scattering angle of $\Theta$ and $R_{\text{mult}}$ is the multiple scattered radiation. Tanaka et al. (1986) used a forward scattering around $\Theta = 20°$ at which the phase function is relatively independent of the SDF of the atmospheric particulate matter. Extending this principle, SKYNET adopts the IL method to extrapolate Eq. (3) regarding the total scattering optical path,

$$x = m_0\omega\tau, \tag{5a}$$

or its aerosol part,

$$x_a = m_0 \omega_a \tau_a, \tag{5b}$$

which can be retrieved from the forward scattering part, $\Theta \leq 30°$, of the relative sky radiance $R$, Eq. (4). The formulae in Eqs. (4) and (5) indicate that $x_a$ is relatively accurately retrieved from the inversion of the forward scattering part of the sky radiance. We use Eq. (5b) in most of ILPs of the sub-networks.

The accuracy of $F_0$ estimation by the IL method depends on the turbidity condition of the site. The theory of a linear regression model is formulated with a normal random observation error $u$ as

$$y_i = a + b x_i + u_i, \quad i = 1, \dots, n, \tag{6a}$$
$$a = \ln(F_0), \quad x = m_0 \omega \tau, \quad y = \ln(F), \tag{6b}$$

where $n$ is the number of observations. Here, we omit subscript $a$ from $\tau_a$ and $\omega_a$ for the sake of compact notation unless otherwise specified. Equation (6) gives estimates of regression coefficients and their dispersion as

$$b = \frac{<(x-\overline{x})(y-\overline{y})>}{\sigma_x^2}, \quad a = \overline{y} - b\overline{x}, \tag{7a}$$

$$\sigma_b^2 = \frac{\varepsilon_u^2}{n\sigma_x^2}, \quad \sigma_a^2 = \frac{\varepsilon_u^2}{n}\left(1 + \frac{\overline{x}^2}{\sigma_x^2}\right), \tag{7b}$$

where upper bar and <> TS8 stand for averaging operation and $\varepsilon_u$ is the root-mean-square error (RMSE) for $u$. The standard linear regression theory assumes $x$ is an independent variable to be related to a dependent variable $y$ that includes a random residual of the fitting $u$. Based on this assumption, the dispersion of $x$ is given as

$$\left(\frac{\sigma_x}{\overline{x}}\right)^2 \sim \frac{\sigma_{m0}^2}{\overline{m}_0^2} + \frac{\sigma_\tau^2}{\overline{\tau}^2} + \frac{\sigma_\omega^2}{\overline{\omega}^2}, \tag{8a}$$

where $\sigma_{m0}^2, \sigma_\tau^2$, and $\sigma_\omega^2$ are dispersions of sampling air masses $\{m_{0i}\}$ and natural variations in $\{\tau_i\}$ and $\{\omega\}$ during the ILP, respectively. The dispersion of residual $\{u_i\}$ is approximated by the sum of mean-square errors of $\tau$ and $\omega$, i.e., $\varepsilon_\tau^2$ and $\varepsilon_\omega^2$, caused by the inversion process of Eq. (4) as

$$\varepsilon_u^2 = b^2 \overline{m}_0^2 \left[(\overline{\omega}\varepsilon_\tau)^2 + (\overline{\tau}\varepsilon_\omega)^2\right] + \varepsilon_F^2, \tag{8b}$$

where $\varepsilon_F^2$ is the mean-square error of $\{y_i\}$ caused by observations of the radiometer, which is usually small and neglected from the formula. The budget of dispersions Eq. (8a) leads to the following estimate for a typical air mass sampling from $m_1 = 1.3$ to $m_2 = 3.5$ and atmospheric conditions of large optical parameter change from $\tau_1 = 0.2$ to $\tau_2 = 0.4$ and from $\omega_1 = 0.85$ to $\omega_2 = 0.95$ during the ILP as

$$\left(\frac{\sigma_x}{\overline{x}}\right)^2 \approx \frac{1}{3}\left(\frac{m_2-m_1}{m_2+m_1}\right)^2 + \frac{1}{3}\left(\frac{\tau_2-\tau_1}{\tau_2+\tau_1}\right)^2$$
$$+ \frac{1}{3}\left(\frac{\omega_2-\omega_1}{\omega_2+\omega_1}\right)^2 = 0.070 + 0.037 + 0.001, \tag{9}$$

if we assume a regular sampling of linear change models for $m$, $\tau$, and $\omega$. This budget indicates that the wide sampling of air mass is the main contributor to decrease $\sigma_a^2$. The IL method allows selection of the atmospheric condition in which $\tau$ and $\omega$ undergo natural variations that help to increase $\sigma_x$ and thus decrease $\sigma_a$. But such selection of unstable atmospheric conditions may increase inversion errors, $\varepsilon_\tau$ and $\varepsilon_\omega$, wasting the benefit of natural changes in $\omega$ and $\tau$. It is also possible to have a change in atmospheric conditions over a short time of less than 5 min in one full angle scan, causing unexpected errors. Sub-networks, therefore, have their own screening protocols for ILP using stability of the time sequence of variables to reject ill conditioned data for ILP. They also reject large AOT cases to secure the calibration accuracy, e.g., AOT > 0.4 by ESR (Campanelli et al., 2004).

Combining Eqs. (7) and (8), we have the following estimate for $\sigma_a$ assuming $b$ and $\omega$ are close to 1,

$$\sigma_{a,\text{IL}} = \frac{3.5}{\sqrt{n}} \frac{\overline{m}_0^2}{|m_2-m_1|} \gamma\overline{\tau} \sim \frac{9.2}{\sqrt{n}}\gamma\overline{\tau} \sim \frac{1.3}{\sqrt{n}}\overline{\tau}, \tag{10a}$$

$$\gamma \equiv \sqrt{\left(\frac{\varepsilon_\tau}{\overline{\tau}}\right)^2 + \left(\frac{\varepsilon_\omega}{\overline{\omega}}\right)^2}. \tag{10b}$$

The third expression on the right-hand side of Eq. (10a) is an estimate for $m_1 = 1.3$ and $m_2 = 3.5$, and the rightmost expression is an approximation with 10 % relative errors in inversion of $\tau$ and $\omega$ as a typical example of ILP. This estimate indicates that the accuracy of $\ln(F_0)$ from the IL method is proportional to the OT during ILP operation at the site. Table 3 lists mean values of $n$, $\tau_a$ and $\sigma_{a,\text{IL}}$ every 30 d (month) obtained by ILP operation carried out at the Tokyo University of Science (TUS) and Rome sites. The table shows that the monthly value of $\sigma_{a,\text{IL}}$ ranges from 0.5 % to 2.4 %, with a tendency to increase with decreasing wavelength. We also estimated $\sigma_{a,\text{IL}}$ by Eq. (10) with optimum $\gamma$ values of 7 % and 15 % for Tokyo and Rome, respectively. These estimates correspond to 5 % and 11 % for relative retrieval errors $\varepsilon_\tau/<\tau>$ and $\varepsilon_\omega/<\omega>$ during ILP operation.

The monitoring ability of $F_0$ by IL on-site methods has merits, such as low-cost frequent calibration to detect the changing constants and a short-term ESI change, and minimizes the radiometer environmental change, avoiding shipping for calibration. The error in $F_0$ is propagated to cause an error in OT from Eq. (3) as

$$\varepsilon_{\text{direct},\tau} \sim \frac{\sigma_a}{m_0}. \tag{11}$$

A rough estimate of AOT error by the IL calibration is expected to be on the order of 0.003 to 0.01 for $m_0 = 2$ in the case of Table 3, though real errors depend on detailed setup and observation sequence at each site. It is important to compare this accuracy of IL with that of SL. In the SL case, we assume $x = m$ in Eq. (6a), so that the error estimate Eq. (7b)

**Table 3. (a)** Monthly mean values of $n$, $\tau_a$, and $\sigma_{a,\mathrm{IL}}$ obtained by ILP at the Tokyo University of Science (TUS) site averaged for the period of February–May 2017 and **(b)** those at the Roman site averaged for October 2017 and May–September 2019, other than the 380 nm data, which were taken only in October 2017. Estimates of $\sigma_{a,\mathrm{IL}}$ are also given by Eq. (10) assuming $\gamma$ value of 7 % for Tokyo and 15 % for Rome. All wavelength means are shown in the bottom of each table.

**(a) Tokyo**

| $\lambda$ (nm) | $<n>$ | $<\tau_a>$ | $<\sigma_{a,\mathrm{IL}}>$ (%) | Eq. (10) (%) $\gamma = 7\%$ |
|---|---|---|---|---|
| 340 | 230 | 0.363 | 1.43 | 1.54 |
| 380 | 189 | 0.343 | 2.37 | 1.61 |
| 400 | 223 | 0.321 | 1.26 | 1.38 |
| 500 | 271 | 0.241 | 1.02 | 0.94 |
| 675 | 257 | 0.159 | 0.77 | 0.64 |
| 870 | 184 | 0.121 | 0.54 | 0.57 |
| 1020 | 202 | 0.100 | 0.47 | 0.45 |
| Mean | 222 | 0.235 | 1.12 | 1.02 |

**(b) Rome**

| $\lambda$ (nm) | $<n>$ | $<\tau_a>$ | $<\sigma_a>$ (%) | Eq. (10) (%) $\gamma = 15\%$ |
|---|---|---|---|---|
| 340 | 360 | 0.178 | 1.25 | 1.29 |
| 380 | 135 | 0.099 | 0.98 | 1.18 |
| 400 | 366 | 0.171 | 1.51 | 1.23 |
| 500 | 360 | 0.120 | 0.87 | 0.87 |
| 675 | 336 | 0.081 | 0.56 | 0.61 |
| 870 | 321 | 0.061 | 0.57 | 0.47 |
| 1020 | 315 | 0.057 | 0.61 | 0.44 |
| Mean | 313 | 0.110 | 0.91 | 0.85 |

is reduced to the following expression as

$$\sigma_{a,\mathrm{SL}}^2 = \frac{\tau'^2}{n}\left(1 + \frac{\overline{m}^2}{\sigma_m^2}\right), \tag{12a}$$

where we assume the error in $a$ is caused by a part of OT change during the SL plot, which tends to the inverse of the optical air mass as

$$\tau = \overline{\tau} + \frac{\tau'}{m}. \tag{12b}$$

A measure of OT change during air mass change from $m_1$ to $m_2$ can be defined as

$$\delta_\tau \equiv \frac{\tau_2 - \tau_1}{2} = \frac{1}{2}\left(\frac{1}{m_2} - \frac{1}{m_1}\right)\tau' = 0.24\tau'. \tag{12c}$$

The rightmost estimate is given for $m_1 = 1.3$ and $m_2 = 2.5$ as an example. If we assume $\delta_\tau/\overline{\tau} = 0.1$ to be the same as the inversion error in the estimate of IL accuracy, the following estimate is given as

$$\sigma_{a,\mathrm{SL}} = \frac{1.6}{\sqrt{n}}\overline{\tau}. \tag{13}$$

This estimate of the SL error is similar to that of IL given in Eq. (10), suggesting the SL performance is similar to or slightly larger than that of IL under conditions of 10 % change in OT during the SL plot. Selection of the calibration methods, therefore, depends on the character of the turbidity conditions at the site. There are reports from city-area sites, such as Rome, Beijing, and Chiba, that the accuracy of SL method is more than 1 % to 2 % lower than that of IL method, suggesting $\varepsilon_\tau/\overline{\tau} > 0.1$ [TS9] commonly happens at these sites, and thus we recommend comparison of $F_0$ values from both SL and IL methods to diagnose the calibration quality of the SL and IL methods. At the same time, we recommend high mountain calibration and/or transfer of calibration constants from a well-calibrated standard radiometer to keep the on-site IL calibration healthy.

The SKYNET community performed high mountain calibrations at Mauna Loa (USA, 3397 m a.m.s.l. [TS10]) and two similar pristine aged-background sites (AOT500 ~ 0.05, AOT at $\lambda = 500$ nm) from the Indian Astronomical Observatory (IAO) at Hanle (Mt. Saraswati, 32°47′ N, 78°58′ E, 4500 m a.m.s.l.) and Merak (33°48′ N, 78°37′ E, 4310 m a.m.s.l.), located in the high-altitude Ladakh region in the northwestern Himalaya. Figure 4 shows retrieved values of $F_0$ and SVA from the observation taken by a single instrument (POM-01) from IAO-Hanle during January 2008–December 2010 and June 2015–December 2018 and Merak during January 2011–May 2015. They used the Skyrad pack software for data screening with a condition of root-mean-square difference (RMSD) of SVAs below 0.20, while the median value of the long-term data is as much as 0.05. The observations were taken from a wide range of AOTs with minimum (instantaneous) 0.01 to maximum 0.22 with the annual averaged AOT as $0.045 \pm 0.026$ at 500 nm during 2008 to 2018 at the two sites. Due to limiting cloudy conditions in the afternoon, 35 % of the disk-scanning work is performed between 08:00 and 09:00 LT [CE10] at this site. Since the disk-scanning procedure takes around 20–25 min to complete the entire wavelengths, it is apparent that in some cases, some wavelengths may have been affected by thin (cirrus) clouds, which are carried by strong winds (above 15 m s⁻¹ [TS11]) at both the sites. The figure indicates that the RMSD of $\ln(F_0)$ from SL and IL methods agree within about 0.5 %. This $F_0$ uncertainty is smaller than the minimum value of $\sigma_{a,\mathrm{IL}}$ [TS12] of about 0.5 % at Tokyo and Rome shown in Table 3 and corresponds to an estimate of Eqs. (10) and (13), assuming the mean AOT at the site is on the order of 0.03 at $\lambda = 500$ nm and $n = 100$. The figure shows that the disk scan method, discussed in the next section, was obtained with monthly mean SVA within 1.5 % for all the spectral channels. The disk scan was performed from observations taken under full clear-sky conditions with minimum of 3–5 d of data in every month (Ningombam et al., 2014). Therefore, there are 12 values of SVA in all the spectral channels in a year. The vertical bar indicates a representative RMSD of monthly means in each year.

**https://doi.org/10.5194/amt-13-1-2020** Atmos. Meas. Tech., 13, 1–24, 2020

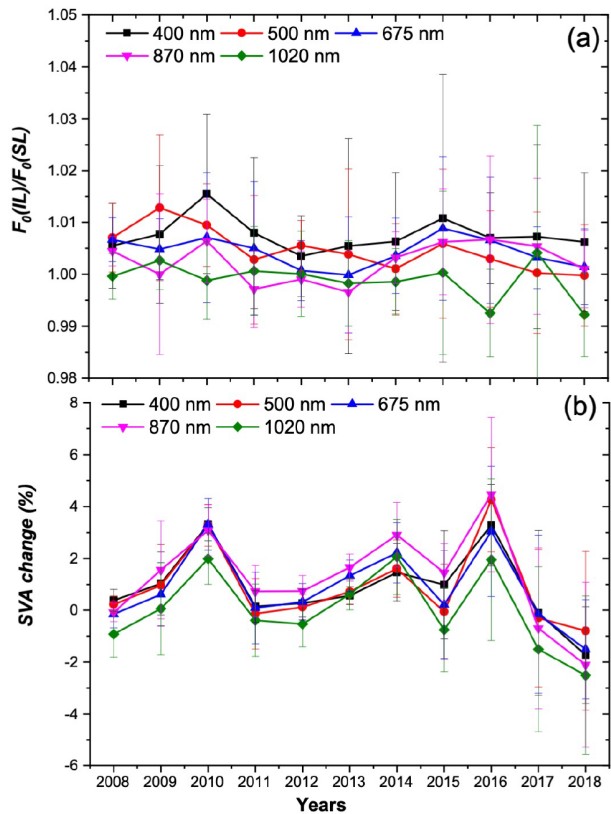

**Figure 4.** Time series of the ratio of $F_0$ values from SL and IL methods **(a)** and SVA **(b)** from the observations taken by a single instrument (POM-01) at two pristine sites, IAO-Hanle during January 2008–December 2010 and June 2015–December 2018 and Merak during January 2011–May 2015. The error bar indicates a representative monthly RMSD in each year.

The first QUAlity and TRaceability of Atmospheric aerosol Measurements (QUATRAM, http://www.euroskyrad.net/quatram.html TS13) Campaign compared the $F_0$ value from the IL method with that of the
5 standard Precision Filter Radiometer (PFR) (Kazadzis et al., 2018b) of the World optical depth Research and Calibration Center (PMOD/WRC). A preliminary analysis showed the difference is 0.3 % at Davos (1590 m a.m.s.l.), where the mean AOT500 is 0.15 and AOT500 in clean aerosol
10 conditions is 0.05. This $F_0$ uncertainty is similar to those of the IOA sites and again smaller than the minimum value in Table 3, indicating the importance of the careful constant calibration effort on high mountains.

Another important point to note is that comparison of Eqs. (3) and (6) leads to the following relation 15

$$b = -\frac{1}{\omega}. \tag{14}$$

The forward scattering analysis of the IL method prescribes the refractive index, and thus it is highly possible for $x$ in Eq. (5a) to include a factor type systematic error like

$$x = Cx_0. \tag{15a}$$ 20

In this case, Eq. (6) results in the following relation between fitted and true values of $a$ and $b$, $a_0$ and $b_0$, as

$$b = \frac{1}{C}b_0, \quad a = \overline{y} - \frac{1}{C}b_0 C\overline{x}_0 = \overline{y} - b_0\overline{x}_0. \tag{15b}$$

This result shows that the formula of $a$ in Eq. (7a) is invariant to the factor type error, indicating the robustness of the IL 25 calibration. On the other hand, the $b$ value changes depending on the value of $C$ and takes a value $-1/\omega$ according to Eq. (14). Boi et al. (1999) utilized this point and proposed an iterative IL method to improve the $F_0$ value and find the optimum CRI by trying several refractive indices. They re- 30 ported the method can improve the precision of $F_0$ by 30 %, e.g., 2 % to 1.5 %.

There is another caution regarding the use of the formulae of Eq. (7a). In the real observation, it is difficult to separate natural variations and inversion errors of $\tau$ and $\omega$, and thus 35 the dispersion $\sigma_x$ tends to include undesired inversion errors that lead the IL method to an underestimation of $a$ and $b$ as understood by Eq. (7b). We are testing a new solution to this problem, named the cross IL method (XIL), which exchanges the role of $x$ and $y$ in the regression analysis, i.e., 40

$$x_i = \alpha + \beta y_i + v_i, \quad i = 1, \dots, n, \tag{16a}$$

$$b = \frac{1}{\beta}, \quad a = -\frac{\alpha}{\beta}. \tag{16b}$$

Figure 5 presents retrieved values of $a$ ($= \ln F_0$) from the IL and XIL methods with 10 ensemble runs of an idealized experiment with $F_0 = 1$; $\omega = 1$; $\tau = 0.1$; $n = 20$; and $m = 1.3$ 45 to 3.5 as a function of normal random errors $\varepsilon_x$ in $x$. The figure shows that the IL method underestimates the $a$ value, while the XIL stays accurate within RMSE less than 0.03 up to $\varepsilon_x = 0.01$ (10 % of $\tau = 0.1$) and 0.05 at $\varepsilon_x = 0.025$ (25 % of $\tau = 0.1$), consistent with Eq. (10). Figure 6 and Table 4 50 compare results of the IL and XIL methods with the following screening conditions applied to 38 sets of real Langley plot data at the TUS site for 4 months from February through May 2017:

$$m_2/m_1 \geq 2, \quad b(\text{SL}) < 10, 0.8 \leq b(\text{IL}), \qquad 55$$

and

$$b(\text{XIL}) \leq 1.2, \quad \varepsilon_u(IL),$$

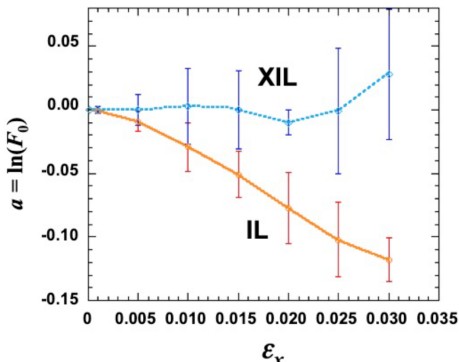

**Figure 5.** Retrieved values of $a = \ln(F_0)$ from IL and XIL methods with 10 ensemble runs of an idealized experiment ($n = 20$ and $m = 1.3$ to 3.5) as a function of normal random error $\varepsilon_x$ in $x$. True values are assumed to be $F_0 = 1$; $\omega = 1$; and $\tau = 0.1$.

and

$$\varepsilon_u(\text{XIL}) \leq \varepsilon_{u0}, \qquad (17)$$

where $m_1$ and $m_2$ are lower and upper limits of air mass in the ILP. The threshold residual $\varepsilon_{u0}$ is given as 0.02, 0.03, and
5 0.05. The figure and table CE11 indicate that the $a$ value from SL is largely scattered, suggesting determination of $F_0$ by SL at turbid sites like Tokyo is not recommendable. On the other hand, $a$ values from IL and XIL converge on a regression line within differences of 2 %–3 %, with a tendency of systemat-
10 ically smaller values by IL than those from the XIL method by amounts of $\varepsilon_{u0}$ and $\varepsilon_{u0}/2$, respectively. Although the difference between IL and XIL is not large when we select low-noise data, we would like to recommend the XIL method to be applied to 5 to 10 Langley plot data sets in order to secure
an accuracy of 1 % to 2 % in $F_0$ using the screening conditions of Eq. (17). The figure also shows that we can detect a long-term decreasing trend of $a$ value by about 10 % during the period at the TUS site.

## 4 Sky radiance calibration for the sky radiometer

Several methods have been proposed for on-site calibration of the sky radiance measured by the sky radiometer, such as the solar disk scan method, point-source or lamp scan method, and diffuse plate method (Nakajima et al., 1986, 1996; Boi et al., 1999). Among them, the solar disk scan
method has been routinely used in the SKYNET measurement of the SVA of the sky radiometer by scanning a circumsolar domain (CSD) of $\pm 1°$ by $\pm 1°$ around the sun at every $0.1°$ interval.

The irradiance received by the radiometer, which is aimed
at the direction $(x, y)$ in a Cartesian coordinate system of angular distance from the center of the solar disk at origin $(x = 0, y = 0)$, is an angular integration of radiances weighted by the response function of the radiometer $f_R$ in

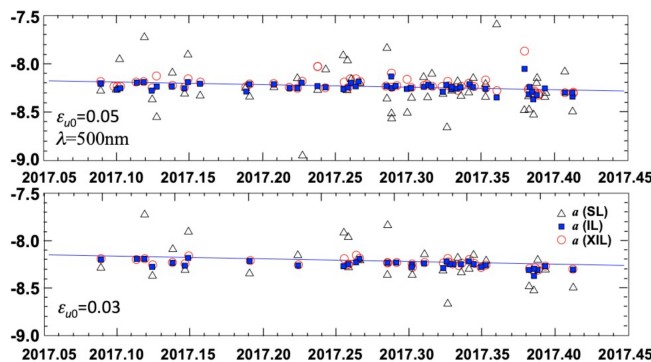

**Figure 6.** Time series of estimated $a$ values by IL and XIL methods for ILP data at the Tokyo University of Science (TUS) site for 4 months from February through May 2017. Presented are the results of two screening conditions of Eq. (17) with $\varepsilon_{u0} = 0.05$ and 0.03 at $\lambda = 500$ nm.

**Table 4.** Estimates of $a$ and $b$ values at $\lambda = 500$ nm and their RMSD values ($\sigma_a$, $\sigma_b$) in the $F_0$ retrieval by IL and XIL methods for ILP data at the Tokyo University of Science (TUS) site for 4 months from February through May 2017. Results of three screening conditions of Eq. (17) with $\varepsilon_{u0} = 0.05$, 0.03, and 0.02 are listed.

| $\varepsilon_u = 0.05$ | | | | |
|---|---|---|---|---|
| Method | $a$ | $\sigma_a$ | $-b$ | $\sigma_b$ |
| SL | $-8.220$ | 0.389 | 0.296 | 0.321 |
| IL | $-8.247$ | 0.050 | 0.968 | 0.082 |
| XIL | $-8.219$ | 0.069 | 1.035 | 0.117 |
| $\varepsilon_u = 0.03$ | | | | |
| Method | $a$ | $\sigma_a$ | $-b$ | $\sigma_b$ |
| SL | $-8.253$ | 0.238 | 0.237 | 0.163 |
| IL | $-8.249$ | 0.039 | 0.973 | 0.070 |
| XIL | $-8.233$ | 0.039 | 1.019 | 0.073 |
| $\varepsilon_u = 0.02$ | | | | |
| Method | $a$ | $\sigma_a$ | $-b$ | $\sigma_b$ |
| SL | $-8.190$ | 0.168 | 0.247 | 0.160 |
| IL | $-8.243$ | 0.030 | 0.990 | 0.064 |
| XIL | $-8.233$ | 0.031 | 1.025 | 0.075 |

the field of view (FOV),

$$F(xy) = \iint\limits_{\text{FOV}} dx'dy' f_R(x' - x, y' - y) L(x', y'). \qquad (18)$$ 35

In the case of diffuse sky radiance measurement, the SVA of the radiometer is given from Eqs. (1) and (18) as

$$\Delta\Omega = \iint\limits_{\text{FOV}} dx'dy' f_R(x', y'). \qquad (19)$$

In the case of the solar disk scan, the main term for $F$ is given as follows under conditions of small contributions from 40

diffuse sky radiation in the CSD,

$$F(xy) = \iint\limits_{\text{FOV}} dx' dy' f_R(x'-x, y'-y) L_d(x', y'), \quad (20a)$$

where $L_d$ is the radiance distribution of the solar disk. The angular aperture of the sky radiometer is about 1°, whereas the solar disk radius is about 0.5°, and thus we can measure the solar disk-averaged value of the radiometer response function as

$$\overline{f}_R(xy) = \frac{F(x, y)}{F_d}. \quad (20b)$$

From Eqs. (1), (20a), and (20b), the following normalization condition has to be fulfilled,

$$\overline{f}_R(0, 0) = 1. \quad (20c)$$

The SVA can be obtained by the angular integration of the radiance in the CSD as follows:

$$I \equiv \iint\limits_{\text{CSD}} dx dy \, \overline{f}_R(xy) = \frac{1}{F_d} \iint\limits_{\text{CSD}} dx dy$$

$$\iint\limits_{\text{FOV}} dx' dy' f_R(x'-x \, y'-y) L_d(x', y')$$

$$= \frac{1}{F_d} \iint\limits_{\text{FOV}} dx' dy' L_d(x', y')$$

$$\iint\limits_{\text{CSD}} dx dy \, f_R(x'-x \, y'-y) = \Delta\Omega. \quad (20d)$$

The last expression is obtained using Eqs. (19) and (20c) if the size of CSD is large enough to include FOV or the contribution outside the CSD is small. These equations indicate that flatness of the response function around the optical axis should be secured in manufacturing the sky radiometer for stable measurement of the direct solar radiation through Eq. (20b). The perfect flatness is realized by optics without an objective lens, which is useful for moving platforms such as aircrafts and ships (Nakajima et al., 1986).

Analyzing data from the solar disk scan, Uchiyama et al. (2018b) found an underestimation of SVA from the disk scan method of 0.5 % to 1.9 % and proposed a correction method by extending CSD size up to a scattering angle of 2.5°, assuming an extrapolation function as illustrated in Fig. 7. They also discussed that the SVA error for the disk scan can exceed 1 % for large AOT conditions such as AOT550 > 0.5 and proposed a subtraction method using sky radiance calculated from the size distribution retrieved from the relative radiance. This subtraction method can reduce the error to 0.5 % for AOT550 < 2 for sky radiance measurements with the minimum scattering angle $\Theta = 3°$. The recent CEReS system has introduced a QC control for setting the optimal value of SVA for each site including Uchiyama's

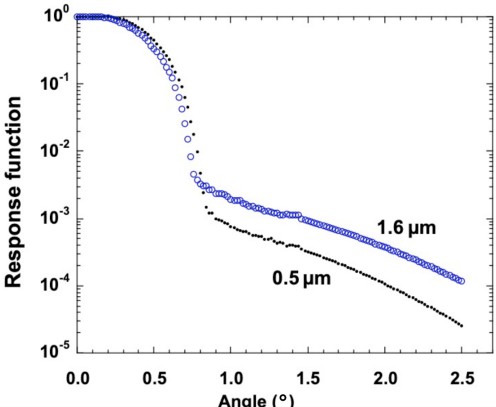

**Figure 7.** Response functions of the sky radiometer at $\lambda = 0.5$ and 1.6 µm measured by the solar disk scan method.

method, but no other sub-networks implement these correction methods in their operational analysis.

Though not performed routinely, a Xe lamp scan has been performed in CEReS for the current version of the sky radiometer (Manago et al., 2016). The merit of the method is that we can narrow the size of the point source below 0.5° and can extend the CSD size beyond ±1° without a significant effect from the sky light. Following this, measured SVAs were compared with those derived from the solar disk scan in daytime. From the experiments, uncertainty in SVA was estimated to be less than ±0.01 msr or ±4 % (Irie et al., 2019). This value is larger than that of Uchiyama et al. (2018b) and more experiments may be needed for more precise estimates and a unit variety.

## 5 Retrievals of parameters for atmospheric constituents

Once the values of radiometer calibration constants, $F_0$, and SVA are determined by the calibration methods described in the preceding two sections, the geophysical parameters of aerosols, clouds, water vapor, and ozone are retrieved by inversion of $F$ and/or $R$ in Eqs. (2a) and (2c) at full or specific scattering angles (Fig. 3). Aerosol retrievals are done using Skyrad pack version 4.2 and/or version 5. The former is based on inversion scheme of the Phillips–Twomey type solution of the first kind of Fredholm integral equation with a homogeneous smoothing constraint, and the latter is based on the second kind of the equation with an inhomogeneous constraint and a priori climate data for aerosols (Twomey, 1963) to retrieve the inherent aerosol optical properties. These methods can be generalized by minimization of a cost function $\varphi$ for realization of an observation vector y as a function of a state vector $\boldsymbol{x}$ TS14 with observation error $\mathbf{e}$ using a multi-term least-squares method (LSM) (Dubovik

and King, 2000; Dubovik, 2004; Dubovik et al., 2011),

$$\mathbf{y} = \mathbf{f}(\boldsymbol{x}) + \mathbf{e}, \tag{21a}$$

$$\phi = \mathbf{e}^{\mathrm{t}} \mathbf{S}_{\varepsilon}^{-1} \mathbf{e} + \phi_1 + \phi_2, \tag{21b}$$

$$\phi_1 = (\boldsymbol{x} - \boldsymbol{x}_{\mathrm{a}})^{\mathrm{t}} \mathbf{S}_{\mathrm{a}}^{-1} (\boldsymbol{x} - \boldsymbol{x}_{\mathrm{a}}), \quad \phi_2 = \boldsymbol{x}^{\mathrm{t}} \mathbf{G} \boldsymbol{x}, \tag{21c}$$

where superscript t stands for matrix transpose operation, $\mathbf{S}_{\mathrm{e}}$ is the error covariance matrix, $\varphi_1$ is the norm of the solution from the a priori data $\boldsymbol{x}_{\mathrm{a}}$ with its associated covariance $\mathbf{S}_{\mathrm{a}}$, and $\varphi_2$ is the cost for smoothness of the solution with the $\mathbf{G}$ matrix related to the norm of the second derivatives of $\boldsymbol{x}$. The AERONET analysis uses both the constraints, $\varphi_1$ and $\varphi_2$, but with only two elements for $\varphi_1$ at the smallest and largest size bins and with the value at the largest size bin as small as is possible to still be able to give a contribution to AOT440 wavelength (Dubovik et al., 2006). Skyrad pack versions 4.2 and 5, respectively, adopt the third and second term of the right-hand side of Eq. (21b), but not both. The latter case of version 5 corresponds to the maximum a posteriori solution (MAP) based on the Bayesian theorem (Rodgers, 2000). The MRI version of Skyrad pack uses a $\varphi_1$ constraint similar to version 5. An iterative search of the nonlinear solution is made by the Gauss–Newton method as

$$\boldsymbol{x}_{i+1} = \boldsymbol{x}_i + \left( \mathbf{K}_i^{\mathrm{t}} \mathbf{S}_{\varepsilon}^{-1} \mathbf{K}_i + \mathbf{S}_{\mathrm{a}}^{-1} + \mathbf{G} \right)^{-1}$$
$$\left[ \mathbf{K}_i^{\mathrm{t}} \mathbf{S}_{\varepsilon}^{-1} (\mathbf{y} - \mathbf{f}(\boldsymbol{x}_i)) - \mathbf{S}_{\mathrm{a}}^{-1} (\boldsymbol{x}_i - \boldsymbol{x}_{\mathrm{a}}) - \mathbf{G} \boldsymbol{x}_i \right], \tag{22a}$$

$$\mathbf{K}_i = \nabla_{\boldsymbol{x}} \mathbf{F}(\boldsymbol{x}) \big|_{\boldsymbol{x} = \boldsymbol{x}_i}. \tag{22b}$$

Version 4.2 uses Eq. (22a) without $S_{\mathrm{a}}$ terms, and version 5 uses the one without $\mathbf{G}$ terms. Observation and state vectors are given as

$$\mathbf{y} = \left\{ \tau_{\mathrm{a}}(\lambda_i), R(\lambda_i \theta_j \phi_j) \, | \, i = 1, \ldots, N_{\lambda}; \right.$$
$$\left. j = 1, \ldots, N_{\mathrm{a}} \right\}, \tag{23a}$$

$$\boldsymbol{x} = \left\{ \ln(V_j), \ln(\tilde{n}_{\mathrm{r}}(\lambda_i)), \ln(\tilde{n}_i(\lambda_i)) \, | \, i = 1, \ldots, N_{\lambda}; \right.$$
$$\left. j = 1, \ldots, N_{\mathrm{v}} \right\}. \tag{23b}$$

where geophysical parameters for the state vector are aerosol volume SDF as a function of logarithm of particle radius $r$, $x = \ln r$, and real and imaginary parts of CRI, i.e., $\tilde{n} = \tilde{n}_{\mathrm{r}} - \tilde{n}_i i$, as functions of wavelength. The SDF is represented by a linear combination of base functions $\{f_k\}$,

$$v(x) \equiv \frac{\mathrm{d}V}{\mathrm{d}x} = \sum_{k=1}^{N_{\mathrm{v}}} V_k f_k(x), \quad x = \ln(r). \tag{24a}$$

The package allows two types of base functions, i.e., box-car functions or lognormal functions with mode radii $\{x_k\}$ that are regularly spaced in $x$ axis,

$$f_k(x) = \frac{1}{\sqrt{2\pi}\sigma} \exp\left[ -\frac{1}{2} \left( \frac{x - x_k}{\sigma} \right)^2 \right]. \tag{24b}$$

The standard analysis in sub-networks assumes 20 lognormal base functions ($N_{\mathrm{v}} = 20$) from $r = 0.02$ to $20\,\mu\mathrm{m}$ with dispersion $\sigma = 0.4$, though there is an argument for a narrower value (Momoi et al., 2020). The a priori value of the CRI is usually given as $\tilde{n} = 1.5 - 0.005i$. The version 4.2 retrieves $x$ through the following four steps: (step 1) the SDF for $x_{\mathrm{a}}$ is assumed to be a bimodal lognormal size distribution ($N_{\mathrm{v}} = 2$) with $r_1 = 0.1\,\mu\mathrm{m}$ and $r_2 = 2\,\mu\mathrm{m}$ and $\sigma_1 = 0.4$, $\sigma_2 = 0.8$, and the volumes of the two modes are set to be the same, $V_1 = V_2$, and are estimated from the forward radiance data ($\Theta \leq 30°$); (step 2) retrieve $\tilde{n}_{\mathrm{r}}$ from the radiance data in $(20, 70°)$; (step 3) retrieve $\{V_k\}$ from the forward radiance data to revise $x_{\mathrm{a}}$; (step 4) retrieve SDF and CRI from the full angle scan data. Step 4 is iterated until a conversion criteria is fulfilled. On the other hand, version 5 follows steps 1, 2, and 4 without step 3. The standard analysis of sub-networks does not treat asphericity of mineral dust and sea salt particles and assumes Mie particles, except for in research studies.

The package adopts the IMS CE12 method for solar aureole radiance calculation (Nakajima and Tanaka, 1988) for full scalar radiative transfer code, Rstar, with polarization correction by Ogawa et al. (1989) to save computing time. A full polarization vector code, Pstar (Ota et al., 2010), is also used for research purposes (e.g., Momoi et al., 2020). Asphericity is treated by an approximation of the method of Pollack and Cuzzi (1980) and by several aspherical kernels. Those softwares are available at OpenCLASTR. Other than polar region measurements, the surface albedo is prefixed at 0.05 or 0.1 for wavelengths shorter than 400 nm and 0.1 at longer wavelengths.

Figure 8a and Table 5 compare observed AOT values with those of AERONET at four co-located sites of Chiba (Japan), Pune (India), Valencia (Spain), and Seoul (South Korea) (Khatri et al., 2016). They found that RMSDs were 0.019 at 675 nm and about 0.015 at 870 and 1020 nm with a site dependence of 0.010, 0.033, 0.009, and 0.022 at 870 nm at the four sites, respectively, though this is not shown in Table 5. Che et al. (2008) compared the AOTs between the POM-02 sky radiometer and Cimel CE-318 sun photometer at the top of the Institute of Atmospheric Physics (IAP) in Beijing, which belongs to SKYNET and AERONET. The POM-02 data were processed by Skyrad pack 4.2. They found an RMSD of 0.025 at 440 nm and 0.018 at other wavelengths, which is similar to the findings of Khatri et al.(2016), even with the mean AOT at this site being as large as 0.33 at 675 nm. RMSDs of the Ångström exponent were 0.19 between 440 and 870 nm and 0.28 between 500 and 870 nm, though this is not shown in Table 5.

SKYNET instruments are regularly compared with Precision Filter Radiometer (PFR) instruments belonging to the World Optical depth Research and Calibration Center (WORCC) and the Global Atmospheric Watch PFR network. Results of three POM instruments compared with the reference WORCC triad in 2015 showed differences of less than 0.005 and 0.01 in all cases and for 500 and 865 nm during the

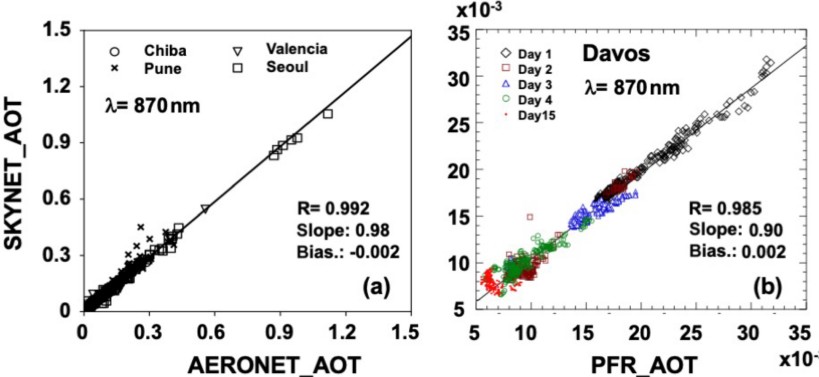

**Figure 8.** Comparison of AOT values at λ = 870 nm obtained by the sky radiometer, Cimel sun photometer, and PMOD PFR.

**Table 5.** Statistics of AOT differences from other networks. RMSD values of Estellés et al. (2012b) are differences between AERONET values and SUNRAD values for the same Cemil-CE318 sun photometer data with mode 1 (SKYNET-like) and mode 2 (AERONET-like) algorithms.

| Source | Statistics | 340 | 380 | 440 | 500 | 675 | 870 | 1020 |
|---|---|---|---|---|---|---|---|---|
| Che et al. (2008), Beijing, | mean | | | 0.536 | – | 0.330 | 0.248 | 0.211 |
| with AERONET | RMSD | | | 0.025 | – | 0.018 | 0.018 | 0.018 |
| Figure 8a, 4 sites,* | mean | | | | | 0.124 | 0.089 | 0.080 |
| Khatri et al. (2016), | RMSD | | | | | 0.019 | 0.015 | 0.016 |
| with AERONET | | | | | | | | |
| Figure 8b, Davos, | mean | | | | 0.041 | – | 0.037 | – |
| Kazadzis et al. (2018a, b), | RMSD | | | | 0.007 | – | 0.001 | – |
| with PFR | | | | | | | | |
| Go et al. (2020), Seoul, | mean | 0.263 | 0.235 | 0.205 | 0.173 | 0.119 | 0.088 | 0.087 |
| with AERONET | RMSD | 0.036 | 0.033 | 0.029 | 0.015 | 0.009 | 0.007 | 0.015 |
| Estellés et al. (2012b), | | | | | | | | |
| Valencia, | | | | | | | | |
| with AERONET (mode 1) | RMSD | 0.018 | 0.013 | 0.011 | 0.010 | 0.010 | 0.008 | 0.010 |
| with AERONET (mode 2) | RMSD | 0.005 | 0.004 | 0.002 | 0.002 | 0.002 | 0.001 | 0.002 |

* These four sites are Chiba, Pune, Valencia, and Seoul.

fourth filter radiometer comparison (Kazadzis et al., 2018a). During the same campaign, Ångström exponent mean differences were less than 0.5. Under low aerosol conditions, a small relative bias in the AOT determination at 500 and 865 nm can theoretically lead to large deviations in the calculated Ångström exponents (AEs). As an example, for AODs of about 0.05 and 0.02 at 500 and 865 nm, respectively, AOT differences of 0.01 and 0.005 can lead to AE differences up to ∼ 1. Since 2015, PFR versus POM long-term comparisons have been performed at various stations, i.e., Valencia (Spain), Chiba (Japan), Davos (Switzerland), and during the QUATRAM campaign in Rome (Italy) (Kazadzis et al., 2018a; personal communication TS15 by Monica Campanelli). Figure 8b and Table 5 compare AOTs at Davos to those of PMOD PFR. The PFR comparison uses the result from the SUNRAD pack (Estellés et al., 2012a), where only direct

measurements from the sky radiometer are used to retrieve AOT, which have a higher time resolution with respect to direct measurements performed during the almucantar scenarios. They found an RMSD as small as 0.007 and 0.001 at 500 and 870 nm, respectively.

Using multi-radiometer observation data since 2016 at Yonsei University, South Korea, in a validation study for the upcoming Geostationary Environment Monitoring Satellite (GEMS) (Kim et al., 2020), Go et al. (2020) compared AOTs from a Cimel sun photometer, Ultraviolet Multifilter Rotating Shadowband Radiometer (UV-MFRSR), NASA Pandora sun spectrometer, and POM-02 sky radiometer. As shown in Table 5, they found RMSDs between AOT values from the POM-02 and Cimel sun photometer of 0.029 to 0.036 for λ ≤ 440 nm and 0.009 to 0.015 for λ ≥ 500 nm.

The statistics shown in Table 5 indicate that the RMSD took a value less than 0.02 for $\lambda \geq 500$ nm and a larger value of about 0.03 for shorter wavelengths in city areas, whereas mountain comparisons show smaller RMSDs of less than 0.01. This location difference can be understood using $F_0$ uncertainties from around 0.5 % to 2.4 % in Tokyo and Rome and smaller values around 0.3 % to 0.5 % at the mountain sites of IOA and Davos, as discussed in Sect. 3, though uncertainties in AOT comparisons can include other error sources, such as pointing error, time variation, and errors in the retrieval software. Estellés et al. (2012b) discussed this point using a comparison of AERONET AOT values with those retrieved by their SUNRAD pack for the same sun photometer, but with two different analysis modes, i.e., mode 1, which implements the SKYNET extinction model, and mode 2 with an AERONET-like model. As listed in Table 5, they found an RMSD of about 0.01 for $\lambda \geq 440$ nm and a larger value in UV channels with a mode 1 setup, whereas a mode 2 setup gives a very small RMSD of less than 0.005. Therefore, more than half of the RMSDs found in the comparison between SKYNET and AERONET can be attributed to differences in the analysis software. The Skyrad pack assumes a simplified extinction model with a plane-parallel assumption in the optical air mass formula, ignores water vapor absorption in IR channels, and has an ozone absorption extinction model in the UV channels that is different from the AERONET model. Slightly larger values at 1020 nm than at 875 nm may be due to the omission of water vapor absorption. Further work is needed to study the effects of these simplifications, which need improvements. For example, SKYNET poses an IL operation limit of $m_0 \leq 3$ instead of $m_0 < 5$ in the data analysis of Estellés et al. (2012b) shown in Table 5.

Table 6 lists reported SSA differences from other networks. SSA values from SKYNET are known to be overestimated, as pointed out by Che et al. (2008). Mean values of SSA in Beijing retrieved from the PREDE sky radiometer were significantly larger than those from the Cimel sun photometer, with differences reaching 0.06 to 0.07 for $\lambda \geq 870$ nm, whereas the mean differences were less than 0.03 at shorter wavelengths. This wavelength dependence can be understood by a tendency for error to increase with decreasing AOT (Dubovik et al., 2000). Similarly Khatri et al. (2016) had a positive difference of about 0.07 RMSD for $\lambda \geq 675$ nm from AERONET values at the four sites (Chiba, Pune, Valencia, and Seoul), and they found that the values can be reduced to around 0.03 if various corrections are applied. The major error source was SVA underestimation of 1.4 % to 3.7 % causing an SSA increase of 0.03 to 0.04. There was an AOT underestimation of 0.02 RMSD at 675 nm, as shown in Table 5, which caused an SSA increase of 0.02 at 675 nm and less than 0.004 at longer wavelengths. Version 4.2 of the Skyrad pack tended to give larger SSA than version 5, but the difference was less than 0.01 for usual aerosol conditions in this case. Effects of surface albedo and asphericity on the SSA difference were less than 0.01.

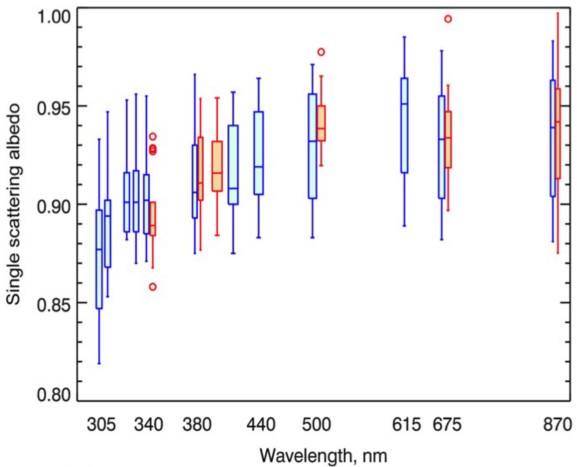

**Figure 9.** Combined spectral SSA from AMP retrievals (blue symbols) and SKYNET retrievals (orange symbols) using MODIS-derived surface albedo. The bottom and top edges of the boxes are located at the sample 25th and 75th percentiles; the whiskers extend to the minimal and maximal values within 1.5 IQR (interquartile range). The outliers are shown in circles. The center horizontal lines are drawn at the median values. The whisker boxes are computed using AOD440 $\geq 0.4$ criteria to correspond the best quality level 2 AERONET data. Cited from Mok et al. (2018).

These effects are consistent with those obtained by sensitivity simulations by Pandithurai et al. (2008) and Hashimoto et al. (2012) in a similar way to those described by Dubovik et al. (2000). Pandithurai et al., found a 5 % error in $F_0$, SVA and 0.5° error in the azimuth angle pointing in SKYNET, which can induce an error of 0.03 in retrieved AOT and mean, and maximum differences in retrieved SSA are about 0.004 and 0.02. Hashimoto et al.(2012), found, in a numerical simulation at 500 nm, as shown in Table 6, a positive SSA retrieval error of $+0.03$ can be caused by SVA underestimation of about 5 %, AOT underestimation of about $-0.02$, and ground albedo underestimation of about $-0.1$.

Aerosol properties in the UV spectral region were extensively measured in the KORUS-AQ campaign (https://espo.nasa.gov/korus-aq/content/KORUS-AQ TS16). Mok et al. (2018) compared SSA retrievals, as shown in Fig. 9 and Table 6, from SKYNET SR-CEReS, AERONET, and Pandora AMP radiometers from April to August 2016 at Yonsei University, South Korea. They found differences of around 0.02 for $\lambda \leq 500$ nm and a larger value of 0.05 at 870 nm, similar to those of Che et al. (2008) and Khatri et al. (2016) shown in Table 6. They also found that the SSA difference increased by 0.004 to 0.008 at short wavelengths when they adopted a spectrally fixed ground albedo $A_g$ at 0.1 as was assumed in the SKYNET analysis, instead of the original setup of spectrally varying AERONET ground albedo $A_g$.

Cloud contamination is another significant error source, as studied by Hashimoto et al. (2012). They studied a case of cirrus contamination detected by a lidar observation in

**Table 6.** Reported SSA differences from other networks: mean bias (Che et al., 2008) and RMSD (Kim et al., 2005; Khatri et al., 2016; Mok et al., 2018). Simulated changes of SSA between Skyrad pack versions 4.2 and 5 and SSA retrieval errors of version 4.2 in an enhanced mineral dust case are also obtained by a numerical simulation (Hashimoto et al., 2012).

| Source | Method | 340 | 380 | 440[1] | 500 | 675 | 870 | 1020 |
|---|---|---|---|---|---|---|---|---|
| Kim et al. (2005), RMSD | Diffuse to direct | | | | 0.027 | | | |
| Che et al. (2008), with AERONET, mean | | | | 0.01 | 0.03 | 0.03 | 0.06 | 0.07 |
| Khatri et al. (2016), 4 sites[2], with AERONET RMSD | Before correction | | | | | 0.069 | 0.074 | 0.068 |
| | After correction | | | | | 0.027 | 0.030 | 0.037 |
| Mok et al. (2018), with AERONET RMSD | Spectral $A_g$ | 0.017 | 0.015 | 0.016 | 0.025 | | 0.047 | |
| | $A_g = 0.1$ | 0.025 | 0.018 | 0.020 | 0.024 | | 0.048 | |
| Hashimoto et al. (2012), simulation mean | Beijing observed cirrus contamination, version 4.2 and 5 | | | | 0.017 | 0.023 | 0.029 | 0.035 | 0.023 |
| difference | Enhanced mineral dust case, version 4.2 | | | | 0.008 | 0.004 | −0.005 | −0.013 | −0.017 |
| | and 5 | | | | −0.013 | −0.017 | −0.026 | −0.031 | −0.030 |

[1] 400 nm in Hashimoto et al. (2012). [2] Chiba, Pune, Valencia, Seoul.

Beijing and found that Skyrad pack version 4.2 retrieved SSA values larger by between 0.017 and 0.035 than those from version 5, as shown in Table 6. Version 4.2 simply retrieves a cloud particle volume as coarse-mode aerosol volume with the smoothness constraint $\varphi_2$ in Eq. (21), but version 5 can filter out the cloud particles, owing to the a priori constraint $\varphi_1$ on SDF. This robustness of version 5 to cloud contamination makes the inversion of the aerosol SDF robust to various noises, as reported by Che et al. (2014) and Jiang et al. (2020), who demonstrated a clear aerosol bimodal size distribution over Beijing in China by using Skyrad pack version 5. Hashimoto et al. (2012), therefore, proposed a data screening protocol to reject unusually large coarse-particle volume: (C1) AOT500 < 0.4, (C2) $\varepsilon_\tau < 0.07$, and (C3) $2 \times V_{2.4\,\mu m} < \max(V_{7.7\,\mu m}, V_{11.3\,\mu m}, V_{16.5\,\mu m})$. Application of this screening protocol reduced SSAs by version 4.2 to be closer to version 5 and AERONET values within 0.03 for 8 to 9 month data at Pune (India) and Beijing (China).

It is also interesting to compare the sky radiometer method with other methods. Kim et al. (2004, 2005) compared SSAs from a sky radiometer with those estimated by the diffuse direct method (King and Herman, 1979 TS17) using data from a co-located pyranometer network in the APEX campaign (Asian Atmospheric Particle Environmental Change Studies) (Nakajima et al., 2003). This method is especially beneficial for the climate study community, because the method gives effective SSA values consistent with the Earth radiation budget. They found an RMSD at 500 nm of about 0.03 from data on Amami Ōshima Island. This value is consistent with other values in Table 6.

One reservation about the SSA retrieval by version 5, though, is that it tends to underestimate the SSA due to underestimation of the coarse aerosols when the a priori SDF for constraint tends to zero for radii larger than 10 μm. Hashimoto et al. (2012) showed by their numerical simulation of an enhanced mineral dust case that version 5 tends to underestimate SSA by 0.017 to 0.035 compared to version 4.2, as shown in Table 6, because version 5 mistakenly filters out coarse aerosols using the a priori SDF data $x_a$ in Eq. (21c). Estellés et al. (2018) found similar underestimation of the coarse aerosols by version 5 compared to aircraft in situ observations (Marenco et al., 2018; Ryder et al., 2018) for African dust events in the sun photometer Airborne Validation Experiment in Dust (SAVEX-D) campaign during 16–25 August 2015, as shown in Fig. 10. The figure indicates that version 4.2 retrieved coarse-mode SDF similar to the observed SDF, though the error bar is large. These examples suggest an improvement of the a priori SDF data is needed for severe dust storm cases.

Water vapor amount is retrieved from direct solar irradiance measurement in the 940 nm channel. $F_0$ value in the water vapor channel is retrieved by the modified Langley plot (ML or MLP) method based on the following OT formula instead of Eq. (3):

$$y = \ln(F_0) - a_g x, \tag{25a}$$

$$y = \ln(F) + m(\tau_a + \tau_R), \quad x = (m_g C_g)^{b_g}, \tag{25b}$$

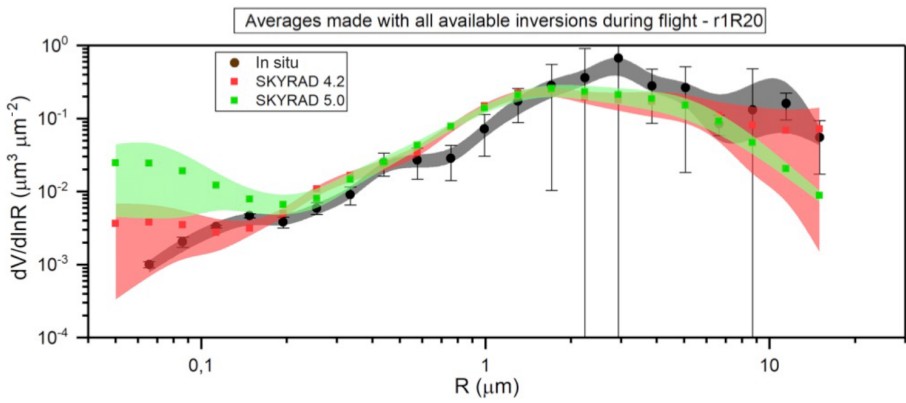

**Figure 10.** Retrieved and observed aerosol size distribution functions in the African dust event cases in the sun photometer Airborne Validation Experiment in Dust (SAVEX-D) campaign during 16–25 August 2015 (Estellés et al., 2018; Marenco et al., 2018; Ryder et al., 2018).

where $\tau_a$ and $\tau_R$ are AOT and OT for molecular scattering, respectively, and $C_g$ is the column-integrated burden of gaseous species, i.e., PWV $W$ in this case; $m_g$ is optical air mass for the gaseous species; $a_g$ and $b_g$ are two prescribed constants to approximate the beam transmittance due to the gaseous species; and $a_g$ can be regarded as an equivalent absorption coefficient for band-averaged absorption of the gaseous species. It is common to assume $m_g$ to be the same as that of atmospheric air mass, i.e., $m_g = m$ in the water vapor case. The value of $\tau_a$ is obtained by an interpolation of the AOT spectrum retrieved from the non-gas absorption channels. There are two algorithms for the SKYNET analysis. One is to use the measured spectral response function of the interference filter of the sky radiometer to prescribe values of $a_g$ and $b_g$ by the theoretical absorption calculation (Uchiyama et al., 2014). This method is similar to that of the AERONET method. The strong line absorption theory of the 930 nm spectral band yields $b_g = 0.5$ (Goody and Yung, 1989) in Eq. (25b). However, there is some dependence of $b_g$ on the vertical structure of the atmosphere, and therefore an improved method is proposed by Campanelli et al. (2010, 2014, 2018) to determine $a_g$ and $b_g$ values using a statistical regression technique of daily observation data at the site. They obtained a range of $b_g$ values of 0.53 to 0.61 as monthly mean values of the 3 years from 2007 to 2009 at the San Pietro Capofiume site (SPC; 44°23′ N, 11°22′ E, 11 m a.m.s.l.), Italy, with some seasonal dependence. One complexity of this method, however, is a need for measurements of $W$ for making the regression analysis. They used PWV either from radiosonde data or a proxy of PWV constructed from surface meteorological data of temperature and relative humidity. Figure 11 compares PWV by the two methods with GPS and AERONET retrievals in Tsukuba, Japan, and Valencia, Spain, for data taken in 2011. The figure CE13 shows that the RMSD from the validation data is less than 0.2 cm using both methods, with some systematic underes-

timation of the slope of the regression line of 10 % in the former method. Estellés et al. (2012b) compared PWV at Valencia, Spain, between AERONET values and those retrieved by the SUNRAD pack for the same Cimel sun photometer. They found an RMSD of 0.20 cm when the SUNRAD pack uses the mode 1 (SKYNET-like) setup, whereas it is reduced to 0.17 cm if SUNRAD uses the mode 2 (AERONET-like) setup, indicating performances of the two modes are similar in water vapor retrievals compared to a significant difference in the AOT case, as shown in Table 5.

In order to get rid of the $F_0$ retrieval process in the water vapor channel, Momoi et al. (2020) proposed a new method of using water vapor dependence of the relative radiance along the almucantar circle of the sky. Although this method has a limited range of retrievable PWV that is less than 2 cm, there is merit in using the value from the method, e.g., $W_{sky}$, as a proxy of $C_g = W$ in Eq. (25b) to perform the MLP on site, similar to the IL method for the non-absorption channels, but with

$$x = W_{sky}^{b_g}, \tag{26}$$

instead of Eq. (25b).

The columnar ozone amount ($O_3$) is retrieved from the direct solar irradiance measurement of 315 nm channel for the Huggins band. Khatri et al. (2014) determined the $F_0$ value using an ML method Eq. (26) assuming $b_g = 1$ for ozone without a significant line absorption structure. The formula of $m_g$ is given by Robinson (1966). In the $F_0$ determination process, they simultaneously obtained an optimal value of the equivalent ozone absorption coefficient $a_g$, which brings the slope of the ML plot to unity using data of ozone column burden $C_g = U$ measured by the Dobson spectrometer. The RMSD of the fitting for campaign data at the Tsukuba site from 13 December 2012 to 8 January 2013 was 13 DU (Dobson unit) as shown in Fig. 12. They also reported a large degradation of filter transmission in the ozone channel.

https://doi.org/10.5194/amt-13-1-2020

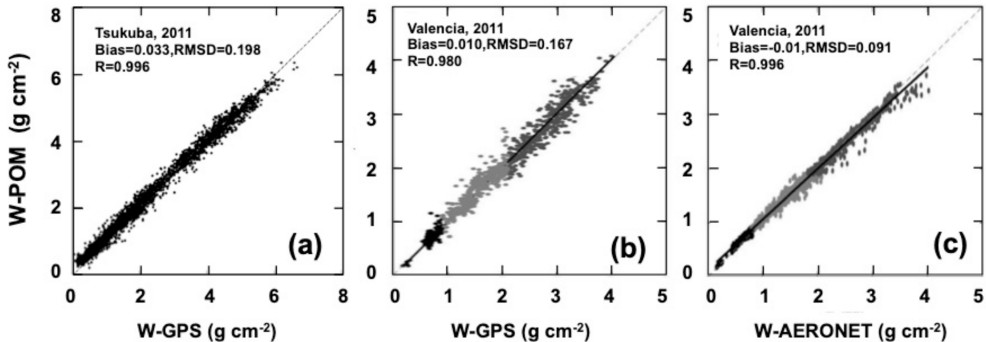

**Figure 11.** Precipitable water retrieved by Uchiyama et al. (2014) in **(a)** and by Campanelli et al. (2018) in **(b)** and **(c)**.

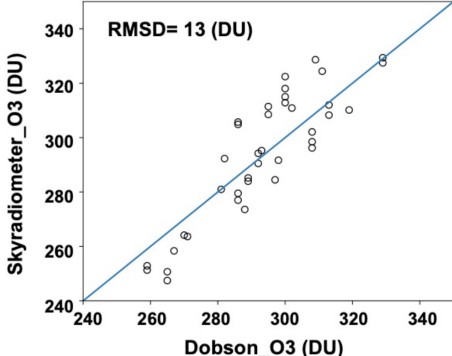

**Figure 12.** Comparison of column ozone amount (DU) retrieved from the sky radiometer at MRI Tsukuba site and from Dobson spectrometer at the JMA Tateno Observatory from 13 December 2012 to 8 January 2013.

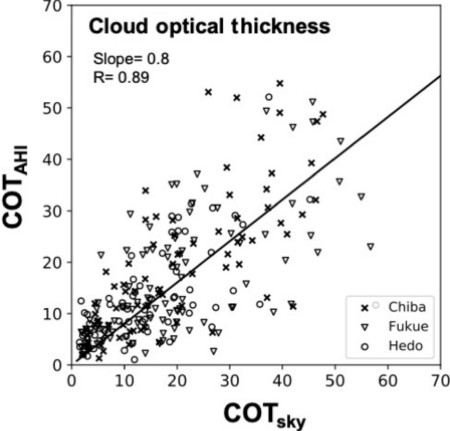

**Figure 13.** Comparison of cloud optical thickness (COT) retrieved from sky radiometer at the Chiba, Fukue, and Hedo sites and the Himawari-8/AHI satellite-borne imager in the period of October 2015 to December 2016 (Khatri et al., 2019). The regression line is shown with zero intercept constraint at the origin.

Cloud microphysical properties have been obtained from diffuse sky radiance measurements from satellites (Nakajima and King, 1990). A similar approach can be applied to the ground-based radiance measurements. Chiu et al. (2010, 2012) retrieved cloud optical thickness (COT) and effective particle radius (CER) from AERONET data. SKYNET uses the POM-02 sky radiometer, which has 1.6 and 2.2 μm channels (Kikuchi et al., 2006; Khatri et al., 2019). Figure 13 compares COT retrieved from POM-02 at zenith observations at the three sites of Chiba, Fukue, and Hedo combined with retrievals from Himawari-8/AHI satellite-borne imager in a period of October 2015 to December 2016 (Khatri et al., 2019). Satellite retrieval results were obtained by the Comprehensive Analysis Program for Cloud Optical Measurement (CAPCOM) (Nakajima and Nakajima, 1995) in the system of AMATERASS (Takenaka et al., 2011; Damiani et al., 2018). Geostationary satellite observation has the merit of frequent time-matching with the ground-based observation. The figure CE14 shows that there is a large scatter of RMSD at 10.2 and a correlation of 0.89. They also studied cloud effective particle radius but did not find a significant correlation between SKYNET and AHI observations. Figure 14

also compares the broadband radiance at zenith measured by a ground-based pyrheliometer and with broadband horizontal radiative flux measured by a pyranometer with those theoretically calculated using the cloud parameters from sky radiometer measurement. The figure CE15 indicates that the down-welling radiance at zenith was consistent between the two radiometers, but horizontal radiative fluxes were not well represented by the cloud optical properties retrieved from the sky radiometer at nadir. Figures 13 and 14 suggest that the inhomogeneity of cloud fields is the main source of differences between the cloud parameters obtained by the sky radiometer and satellite measurements.

# 6 Conclusions

The SKYNET community has undertaken efforts at improving their on-site calibration and analysis systems to provide retrieved aerosol and other atmospheric constituents.

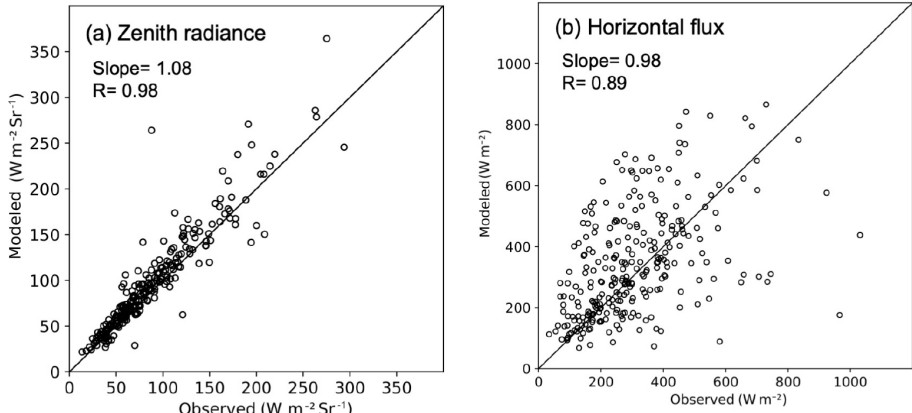

**Figure 14.** The same as in Fig. 13 but for the comparison between modeled and observed broadband **(a)** radiances and **(b)** horizontal radiative fluxes. Regression lines are shown with zero intercept constraint at the origin.

An estimate of the retrieval accuracy of $F_0$ is given by Eq. (10) for the IL method, which can serve as an approximation of observed monthly mean uncertainty in $F_0$ as 0.5 % to 2.4 % at the Tokyo and Rome sites and smaller values of around 0.3 % to 0.5 % at the mountain sites of IOA and Davos. These values are consistent with RMSD values, in the AOT comparisons with other networks, of less than 0.02 for $\lambda \geq 500$ nm and a larger value of about 0.03 for shorter wavelengths in city areas and smaller values of less than 0.01 in mountain comparisons. We also developed a new XIL method to correct an underestimation by the IL method in the case of large aerosol retrieval errors.

Several causes of larger SSA values reaching 0.07 than those of other networks have been identified as underestimation of SVA measured by the disk scan method and the new lamp scan method, cloud contamination, and others. Recent reported values of the difference are found to be less than 0.03 after these corrections.

Retrievals of other atmospheric constituents by the sky radiometer are also reviewed. We found accuracies of about 0.2 cm for the precipitable water vapor amount and 13 DU for the column ozone amount. A new on-site calibration method for water vapor has been developed. The cloud optical properties were found to have some (but not large) correlation with satellite remote sensing values, suggesting cloud inhomogeneity may be one source of error.

There are several aims for the next step of the SKYNET to make its system more reliable and useful for the science community. The reported useful improvements of the product quality are still in the research phase, and it is important to introduce them into the existing operational systems and future system of the ISDC. Comparison studies also showed that the analysis software Skyrad pack may need improvements in its simplified optical model. We want to pursue our on-site calibration system for sustainable operation of the network. However, it is still required for a full accuracy assessment to conduct continuous comparison of on-site calibrations of our standard sky radiometer with high mountain calibrations and other network calibrations.

*Data availability.* .TS18

*Author contributions.* .TS19

*Competing interests.* The authors declare that they have no conflict of interest.TS20

*Special issue statement.* This article is part of the special issue "SKYNET – the international network for aerosol, clouds, and solar radiation studies and their applications (AMT/ACP inter-journal SI)". It is not associated with a conference.TS21

*Acknowledgements.* We are grateful for the support from the JAXA SGLI, MOEJ-JAXA GOSAT and GOSAT2, ERCA/ERDF/S-12, JST/CREST/TEEDDA (JPMJCR15K4) projects. K. MiuraTS22 of Tokyo University of Science is gratefully acknowledged for providing us with Langley plot data. A group of the co-authors were supported by ERCA/ERDF/2-1901, JSPS/KAKENHI/JP19H04235, P17K00529, and the JAXA 2nd research announcement on the Earth Observations (grant number 19RT000351).

ESR is grateful for the support of the Spanish Ministry of Economy and Competitiveness and European Regional Development Fund through funding to the University of Valencia within several projects, such as CGL2015-70432-R and CGL2017-86966-R. ESR also thanks CostantiniTS23 of the Research Area of Tor Vergata for the data center and server maintenance. The SAVEX-D campaign was funded by EUFAR TNA (European Union Seventh Framework Programme, grant agreement no. 312609), making use of airborne data obtained using the BAe-146-301 Atmospheric Research Aircraft operated by Airtask Ltd and managed by the Facility for Airborne Atmospheric Measurements (FAAM). It was a success thanks

to many staff at the Met Office, the University of Leeds, Manchester and Hertsfordshire CE16, FAAM, Directflight Ltd, Avalon Engineering and BAE Systems.

Jhoon Kim was supported by the "Technology development for Practical Applications of Multi-Satellite data to maritime issues" project, funded by the Ministry of Ocean and Fisheries, South Korea.

We thank Alexander Smirnov of NASA GSFC for useful discussions about the history of sun and sky measurements.

*Financial support.* This research has been supported by the Japan Aerospace Exploration Agency (grant nos. SGLI and 19RT000351), the MOEG-JAXA (GOSAT and GOSAT2), the ERCA (grant nos. S-12 and 2-1901), the Japan Science and Technology Agency (grant no. JPMJCR15K4), the Japan Society for the Promotion of Science (grant nos. JP19H04235 and P17K00529), the Spanish Ministry of Economy and Competitiveness and European Regional Development Fund (grant nos. CGL2015-70432-R and CGL2017-86966-R), and the European Union Seventh Framework Programme (grant no. 312609). TS24

*Review statement.* This paper was edited by Omar Torres and reviewed by two anonymous referees.

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

## Remarks from the language copy-editor

## Remarks from the typesetter

**TS18**    Please provide a statement on how your underlying research data can be accessed. If the data are not publicly accessible, a detailed explanation of why this is the case is required. The best way to provide access to data is by depositing them (as well as related metadata) in reliable public data repositories, assigning digital object identifiers (DOIs), and properly citing data sets as individual contributions. Please indicate if different data sets are deposited in different repositories or if data from a third party were used. If no DOI is available, assets can be linked through persistent URLs to the data set itself (not to the repositories' home page). This is not seen as best practice and the persistence of the URL must be secured.

TS25   Please provide page range or article number with DOI.
TS26   Please provide page range or article number with DOI.
TS27   Please provide page range or article number.
TS28   Please provide DOI.
TS29   Please note that this reference has been updated to the final version.
TS30   Please provide all author names and make sure that all authors are listed in the correct order: last name, initial(s).
TS31   Please provide volume number and page range or article number.
TS32   Please check article number and provide DOI.