# Peer review of "An overview and issues of the sky radiometer technology and SKYNET"

_Atmospheric Measurement Techniques, 2020_

## Short Comment (SC1) · 19 Mar 2020

The statement in the Introduction "Combined analyses of sun and sky radiation data were not attained until the 1980s . . ." is not exact. Aureole measurements combined with the direct sun measurements to study atmospheric optical properties and stability were made by Abbot (I do not have a reference though), Kalitin (1930), Fesenkov (1933), Pyaskovskaya-Fesenkova (1957), Bullrich (1964), Lifshits (1965), and Murai (1967). B.N.Holben et al. (2001, Table 1) provided an exhausted history of the long-term optical depth measurements by various researchers over different parts of the world. A nice chronological essay regarding history of the direct sun measurements was presented by G.E.Shaw (2006).

[Figure]

References. Bullrich, K., Scattered radiation in the atmosphere and the natural aerosol, Advances in Geophysics v.10, 99-260, 1964 (https://doi.org/10.1016/S0065-2687(08)60007-2). Fesenkov, V.G., To the question of solar constant determination, Sov. Astron. J., 10(3), 249– 266, 1933 (in Russian). Holben, B.N., et al., An emerging ground-based aerosol climatology: Aerosol optical depth from AERONET, J. Geophys. Res., 106, 12,067-12,097, 2001. Kalitin, N. N., To the question of studying sky radiation intensity around the Sun, Bulletin of Constant Actinometric Commission of Main Geophysical Observatory, 1, 51-56, 1930 (in Russian). Lifshits, G.Sh., Light scattering in the atmosphere, 177 pp., Nauka, Alma-Ata, 1965 (in Russian). Murai, K., Spectral measurements of direct solar radiation and of Sun's aureole (I), Papers in Meteorology and Geophysics, v.18, N3, 239-291, 1967. Pyaskovskaya-Fesenkova, E. V., Investigation of light scattering in the Earth's atmosphere, USSR Acad. of Sciences Press, 218 pp., Moscow, 1957 (in Russian). Shaw, G.E., Genesis of sun photometry, Remote Sensing of Clouds and the Atmosphere XI, edited by James R. Slusser, Klaus Schäfer, Adolfo Comerón, Proc. of SPIE, Vol. 6362, doi: 10.1117/12.692771, 2006.

Please also note the supplement to this comment:
https://www.atmos-meas-tech-discuss.net/amt-2020-72/amt-2020-72-SC1-supplement.pdf
* * *

---

## Referee Comment (RC1) · Anonymous Referee #1 · 12 Apr 2020

" An overview and issues of the sky radiometer technology and SKYNET "

by Nakajima et al.

This paper presents in depth overview of SKYNET network of sun-photometers. It describes the hardware and many details of the network operation including calibration procedures, maintenance, atmospheric aerosol and gaseous property retrievals, as well as validation of the SKYNET products. The SKYNET has been founded about two decades ago and has been dynamically evolved since. In my opinion, the SKYNET together with AERONET is one of the bests established ground-based networks that provided extremely valuable information for validation of satellite observation and directly for aerosol science. It is difficult to overestimate the importance of this information provided by the ground-based networks for current understanding of properties of

atmospheric aerosol and its impact on climate and environment. With no doubts this paper comprising many details of SKYNET operations is clearly interesting and important for the community and for the reader of Atmospheric and Measurement Techniques (AMT). Therefore, I think the paper should be published AMT and included AMT highlights. At the same time, the authors could try to clarify additionally certain aspects and improve the content of the paper even further. Below, I listed few suggestions for optional consideration by the authors.

Detailed comments;

1. The abstract seems to be unusually short, probably it could be extended by adding some more essential information;

2. In my opinion, the paper could be even more interesting if the authors put additional efforts in outlining even more the similarities and differences, as well advantages and disadvantages of SKYNET observations and retrieval products with those from other networks, first of all compare to AERONET.

3. The paper seems to focus on the details of hardware description and acquisition of measurements. Probably, some more information about retrieval procedures could interest the readers.

4. Some statements about accuracy of the retrieval, e.g. about size distribution and single scattering albedo are not justified in fully convincing way. The author just showed few figures and short explanations to them. The justification of retrieval products accuracy normally deserves more attention. For example, in AERONET activities many theoretical investigations and field campaigns are devoted to clarification of the retrieval accuracy of aerosol properties (SSA, etc.). I believe some more discussion and references to filed experiment and numerical tests could be beneficial for readers.

---

## Referee Comment (RC2) · Anonymous Referee #2 · 14 Apr 2020

Review for Atmospheric Measurement Techniques

Title: An overview and issues of the sky radiometer technology and SKYNET

Authors: Teruyuki Nakajima, Monica Campanelli, Huizheng Che, Victor Estellés, Hitoshi Irie, Sang-Woo Kim, Jhoon Kim, Dong Liu, Tomoaki Nishizawa, Govindan Pandithurai, Vijay Kumar Soni, Boossarasiri Thana, Nas-Urt Tugjsurn, Kazuma Aoki, Makiko, Hashimoto, Akiko Higurashi, Stelios Kazadzis, Pradeep Khatri, Natalia Kouremeti, Rei Kudo, Franco Marenco, Masahiro, Momoi, Shantikumar S. Ningombam, Claire L. Ryder, Akihiro Uchiyama

General Comments: A paper focused on the issues and algorithms of the SKYNET network is needed and could prove quite useful to the scientific community. The title

and Introduction section of this paper holds significant promise, however upon reading there was a strong emphasis on calibration methods within SKYNET (which is good), but then a lack of detail or even complete omission of some other important issues. On page 2 (lines 24-26) of the introduction you say: "Compared to the AERONET technology, SKYNET has several differences in measurement and analysis methods, which are useful to overview and assess for the world community to understand the system, which is the purpose of this paper." This sentence outlines the basis for an interesting and useful manuscript, however the differences (between SKYNET and AERONET) in measurement sequences, cloud screening, data quality checking, and some algorithms were not adequately addressed in the current paper. However, revisions with additional in-depth discussion of these issues could make this paper a very valuable and important reference for SKYNET network data users and aerosol researchers worldwide.

Specific Comments:

Abstract: This is likely the shortest abstract I have ever seen, and hardly informative. I recommend that a few specifics be mentioned as there is really nothing of substance in this abstract.

Page 3, lines 4-5: Here you discuss the nations that contribute to SKYNET and that a committee was formed for collaboration. Please add information on the availability of data to the scientific community from these regions/sub-networks and include specific data access websites and data policies.

Page 3, lines 4-5: It is necessary (here or elsewhere in the text) to include information on the temporal frequency of the direct sun and sky radiance measurements for SKYNET sun-sky radiometers. Also important are the types of sky scans made by SKYNET instruments. Are they all almucantar and/or some additional solar principal plane scans? How often are the sky scans made, and over what solar zenith angle (SZA) range? This should be contrasted with the AERONET measurement protocols,

almucantar and hybrid scans. Also over what SZA range are SKYNET almucantars identified as high quality retrievals. For AERONET the minimum SZA for Level 2 (high quality) almucantar retrievals is 50 degrees, corresponding to 100 degrees scattering angle range.

Page 3, lines 4-5: It is surprising that the interference filters for the UV wavelengths would have such a wide band-pass of 10 nm. AERONET filters in the UV have 2 nm band-pass due to solar zenith angle dependent Rayleigh OD issues in measurement of AOD in the UV region.

Page 4, lines 17-18: A short summary of the SKYNET cloud screening is warranted here in the text. It would be very useful to compare what aspects of the cloud screening are similar and which ones differ from the AERONET cloud screening (please refer to V3 cloud screening, discussed in Giles et al (2019)). For example, AERONET uses the angular steepness of the solar aureole as a cirrus filter, while I think that SKYNET does not.

Page 5, line 16: Regarding the improved Langley (IL) method, please provide some estimated ranges of accuracy for different optical depth magnitudes. Also provide accuracy estimates of Fo determination for temporal variation in AOD over the Langley measurement sequence, as it would be expected that this would also be a factor.

Page 6, line 13: Should be 'wasting' instead of 'waisting' here.

Page 6, line 20: Since the AOD is much larger (and more variable) at 400 nm than 870 nm it seems that the error would be larger for the shorter wavelengths. Please explain why this would not be a factor or else include a statement that errors would be larger at shorter wavelengths unless the aerosol is coarse mode dominated (Angstrom Exponent of ∼zero).

Page 7-8 (last 2 sentences of P7 and first 2 sentences of P8): This data in Figure 4b should be discussed in more detail in the next section of Sky radiance calibration

(Section 4).

Section 4, starting on Page 9: Please provide some explanation of the causes of the temporal variance of the SVA as shown in the time series in Figure 4b. This data is from high altitude sites with stable very low AOD and therefore would be expected to be a best case scenario. Also discuss what the variability in SVA from sky scans would look like at a sea level site with moderate and variable turbidity, say AOD=0.5 +-0.3 at 500 nm with alpha=1.5.

Page 10, lines 13-15: It would be useful to provide the reader with an estimate of the magnitude of the error in sky radiance calibration as a result of this uncertainty in SVA from the disk scan. Or else state that the entire data base of SKYNET has been reprocessed with the correction method that was outlined by Uchiyama et al. (2018b).

Page 10, lines 20-24: Why are the retrievals within SKYNET made with two different versions of the Skyrad.pak code (Version 4.2 or version 5)? What are the effects of these different codes on the inverted parameters? Why is there no standardized inversion code for the entire network? Or if I have mis-interpreted this then please clarify in the text what the consistent data processing is for the data from all sites in the SKYNET network.

Page 11, line 1: Please state here whether this Cimel instrument in Beijing is part of the AERONET network with their data processing algorithms or a part of CARSNET network with somewhat different algorithms (especially when compared to V3 of AERONET).

Page 11, lines 2-3: Please explain here why the error in AOD is smaller in the shorter wavelengths even though the AOD and its absolute temporal variance is larger at the shorter wavelengths. This seems to be counter-intuitive, and I would need some significant evidence presented in the paper in order to be convincing.

Page 11, lines 10: This is a relatively large difference in Angstrom Exponent (0.5), but

not if AOD is very low. Please state the AOD levels for these AE comparisons.

Page 11, lines 13: It would be useful to also show or discuss comparison statistics of AOT for 380 nm (in addition to 870 nm in Figure 8) since all instrument types have larger uncertainties at this wavelength.

Page 11, lines 17-18: When discussing differences in AOD it is much more useful to provide the differences in AOD rather than percentages.

Page 11, lines 19: Provide some quantification here of the SSA overestimate found by Che et al. (2008).

Page 11, lines 23: There needs to be more discussion of the SKYNET cloud screening algorithms and comparison to co-located AERONET site data in order for users to understand the cloud contamination issue better.

Page 11, lines 23-25: It is puzzling as to why all the data are not processed with Version 5. This suggests that data at some sites that process the data with V4.2 will have more cloud contamination than other sites that use V5. Please provide some clarification on which versions of SKYRAD.pak software/algorithms are used to process the data for what sites.

Page 11, line 28: More details on the stricter cloud screening in V5 are needed here in the text, not just providing literature references.

Page 12, lines 1-3: Your discussion of SSA differences (Fig 9) is misleading since the agreement was much poorer with standard SKYNET retrievals of SSA. The study of Mok et al. (2018) replaced the crude surface reflectance assumptions utilized by SKYNET with values derived from MODIS measurements (these were AERONET values). More discussion of the spectral surface reflectance used as input to the SKYNET retrievals is needed here.

Page 12-13, last 2 lines of P12 & first 4 line of P13: Please note whether this is a SKYNET retrieval product for W (column water vapor) or just a possibility for future

implementation. Also does SKYNET provide W retrieved from both methods shown in Fig 11 in the database for all sites?

Page 13, line 11: It is very surprising that you use a 10 nm wide filter at 315 nm as stated on page 4 in this paper. This should cause significant SZA dependence in the signal and result in SZA dependence of the retrieved columnar ozone.

Page 13, lines 26-28: With only 1% of the variance explained in this scatterplot (Fig 13b), it seems that there are likely other issues involved in the retrieval of CER from both the sky radiometers and also the satellite measurements. The result could not be any worse than this and therefore it seems like this could just be mentioned in the text without the need of a plot.

Page 14, line 2: This statement is confusing. The accuracy shown in the paper is better than 10% if the AOT is high. For example, if the AOD at a site averages 0.5 for a month then the accuracy of the IL calibration does not result in 0.05 uncertainty.

Page 14, lines 2-3: This 'conclusion' is also misleading since the 'accuracy' of SSA is very hard to define as there is no gold standard for SSA measurement to use as a benchmark. Additionally. as discussed before this value of 0.015 (difference with AERONET and Pandora) is not based on standard SKYNET retrievals but on retrievals made with much more accurate inputs of surface reflectance (AERONET values were used). Also, Mok et al. (2018) applied additional improved quality checks to SKYNET retrievals, not the standard SKYNET product. The way it is currently written would give most readers a false sense of the consistency of the SKYNET retrievals of SSA versus AERONET.

Page 14, line 8: This is the first time the size of the financial budget for SKYNET has been mentioned in the entire paper. It is therefore a bit strange and confusing to include budget considerations in the Conclusions section.

Page 14, lines 10-13: It is odd to put these alternate calibration methods in bullet form

in the Conclusions section. Please format into a typical sentence structure.

---

## Author Comment (AC1) · 9 Jun 2020

24 May 2020

Responses to the comments from referees and open discussion comments

Dear editors, referees and commentator,

Thank you very much for useful comments. We have made a large effort to improve our manuscript following your comments as specified each by each as follows. We believe the manuscript has significantly improved and is informative to the science community.

Teruyuki Nakajima on behalf of the co-authors

**3. Open discussion comments by Dr. A. Smirnov**

C3-1. The statement in the Introduction "Combined analyses of sun and sky radiation data were not attained until the 1980s . . ." is not exact. Aureole measurements combined with the direct sun measurements to study atmospheric optical properties and stability were made by Abbot (I do not have a reference though), Kalitin (1930), Fesenkov (1933), Pyaskovskaya-Fesenkova (1957), Bullrich (1964), Lifshits (1965), and Murai (1967). B.N.Holben et al. (2001, Table 1) provided an exhausted history of the long- term optical depth measurements by various researchers over different parts of the world. A nice chronological essay regarding history of the direct sun measurements was presented by G.E.Shaw (2006).

A3-1. Thank you for an important comment regarding the early-day activities of sun and sky observation. We revised the texts for dating and added description of milestones by Abbot (1911), Abbot and Aldrich (1916), de Bary (1964), Bullrich (1964), Bullrich et al. (1967, 1968), Fesenkov (1933), Gorodetskiy et al. (1976), Holben et al. (2001), Kalitin (1930), Murai (1967), Phillips (1962), Pyaskovskaya-Fesenkova (1957), Roosen et al. (1973), Shaw (2006), Shifrin et al. (1972, 1974), Terez and Terez (2003), Turchin and Nozik (1969), Twitty et al. (1976), Yamamoto and Tanaka (1969).
We also added the name of Dr. Smirnov to the acknowledgments section.

**4. Other revisions/corrections**
The following revisions/corrections were applied other than comments by reviewers.
4-1. We added two authors, Akihiro Yamazaki and Sujung Go, because we needed their data and discussion for producing answers to some of reviewers' comments.
4-2. We added Ministry of Ocean and Fisheries, Korea in Acknowledgments.
4-3. Accuracy of precipitable water vapor was 0.2 cm instead of typo 2 cm in the original manuscript.

4-4. Table numbering changed because of insertion of new Table 3; accordingly Table 3 became Table 4. We also added new Tables 5 and 6. Equation numbering also changed due to insertion of new Eq. (21)-(24).

4-5. We replaced Fig.8(b) from Roman result to Davos result, because, in the new corona virus lock-down in Rome, we had a difficulty to retrieve detailed data from the Italian university where the data are stored.

End of document

---

## Author Comment (AC2) · 9 Jun 2020

Responses to the comments from referees and open discussion comments

Dear editors, referees and commentator,

Thank you very much for useful comments. We have made a large effort to improve our manuscript following your comments as specified each by each as follows. We believe the manuscript has significantly improved and is informative to the science community.

Teruyuki Nakajima on behalf of the co-authors

**1. Referee-1 comments**

C1-1. General comment: This paper presents in depth overview of SKYNET network of sun-photometers. It describes the hardware and many details of the network operation including calibration procedures, maintenance, atmospheric aerosol and gaseous property retrievals, as well as validation of the SKYNET products. The SKYNET has been founded about two decades ago and has been dynamically evolved since. In my opinion, the SKYNET together with AERONET is one of the bests established ground-based networks that provided extremely valuable information for validation of satellite observation and directly for aerosol science. It is difficult to overestimate the importance of this information provided by the ground-based networks for current understanding of properties of atmospheric aerosol and its impact on climate and environment. With no doubts this paper comprising many details of SKYNET operations is clearly interesting and important for the community and for the reader of Atmospheric and Measurement Techniques (AMT). Therefore, I think the paper should be published AMT and included AMT highlights. At the same time, the authors could try to clarify additionally certain aspects and improve the content of the paper even further. Below, I listed few suggestions for optional consideration by the authors.

A1-1. We added more information for improving the content of the paper as explained in answers to each comment.

Detailed comments;

C1-2. The abstract seems to be unusually short, probably it could be extended by adding some more essential information.

A1-2. We extended the abstract summarizing the results of the paper.

C1-3. In my opinion, the paper could be even more interesting if the authors put additional efforts in outlining even more the similarities and differences, as well advantages and

disadvantages of SKYNET observations and retrieval products with those from other networks, first of all compare to AERONET.

A1-3. We tried to put more comparisons in Sections 2 and 5 by adding measurement protocols, QC screening processes, and analysis algorithms with some comparison with those of AERONET to our knowledge. We have added new tables 3, 5, and 6 to quantify the discussion for calibration, AOT comparison, and SSA comparison.

C1-4. The paper seems to focus on the details of hardware description and acquisition of measurements. Probably, some more information about retrieval procedures could interest the readers.

A1-4. We added text in Section 5 to explain algorithms and performance of the analysis system with some comparison with those of AERONET.

C1-5. Some statements about accuracy of the retrieval, e.g. about size distribution and single scattering albedo are not justified in fully convincing way. The author just showed few figures and short explanations to them. The justification of retrieval products accuracy normally deserves more attention. For example, in AERONET activities many theoretical investigations and field campaigns are devoted to clarification of the retrieval accuracy of aerosol properties (SSA, etc.). I believe some more discussion and references to filed experiment and numerical tests could be beneficial for readers.

A1-5. The network is for research purpose without a centralized data analysis system and information is scattered in independent papers and documents, which makes SKYNET difficult to be understood by the science community. In this situation this paper intends to make an overview of key findings and issues of the SKYNET, providing better information for the community. We firstly stated this point in abstract and introduction. We also feel the original manuscript lacks some important details of the software system and data analysis protocols, so that we added more explanations in Section 2 and 5.

   We eliminated 0.015 as the SSA accuracy, and instead added a statement that the difference is less than 0.03 if the improvements are introduced in the operational system, mentioning uses of AERONET knowledge. We also added discussion regarding AOT and SSA accuracies with new Tables 3, 5, and 6. We also added a new Table 2 to list information regarding the known SKYNET data archives.

**4. Other revisions/corrections**
The following revisions/corrections were applied other than comments by reviewers.
4-1. We added two authors, Akihiro Yamazaki and Sujung Go, because we needed their data

and discussion for producing answers to some of reviewers' comments.

4-2. We added Ministry of Ocean and Fisheries, Korea in Acknowledgments.

4-3. Accuracy of precipitable water vapor was 0.2 cm instead of typo 2 cm in the original manuscript.

4-4. Table numbering changed because of insertion of new Table 3; accordingly Table 3 became Table 4. We also added new Tables 5 and 6. Equation numbering also changed due to insertion of new Eq. (21)-(24).

4-5. We replaced Fig.8(b) from Roman result to Davos result, because, in the new corona virus lock-down in Rome, we had a difficulty to retrieve detailed data from the Italian university where the data are stored.

End of document

---

## Author Comment (AC3) · 9 Jun 2020

Responses to the comments from referees and open discussion comments

Dear editors, referees and commentator,

Thank you very much for useful comments. We have made a large effort to improve our manuscript following your comments as specified each by each as follows. We believe the manuscript has significantly improved and is informative to the science community.

Teruyuki Nakajima on behalf of the co-authors

**2. Referee-2 comments**

C2-1. General Comments: A paper focused on the issues and algorithms of the SKYNET network is needed and could prove quite useful to the scientific community. The title and Introduction section of this paper holds significant promise, however upon reading there was a strong emphasis on calibration methods within SKYNET (which is good), but then a lack of detail or even complete omission of some other important issues. On page 2 (lines 24-26) of the introduction you say: "Compared to the AERONET technology, SKYNET has several differences in measurement and analysis methods, which are useful to overview and assess for the world community to understand the system, which is the purpose of this paper." This sentence outlines the basis for an interesting and useful manuscript, however the differences (between SKYNET and AERONET) in measurement sequences, cloud screening, data quality checking, and some algorithms were not adequately addressed in the current paper. However, revisions with additional in-depth discussion of these issues could make this paper a very valuable and important reference for SKYNET network data users and aerosol researchers worldwide.

A2-1. In section 5, we added text regarding the details of the analysis software with some comparison with those of AERONET. And we added new Tables 3, 5 and 6 to quantify the discussion for calibration, AOT comparison, and SSA comparison and mentioning uses of AERONET knowledge. We also added a new Table 2 to list information regarding the known SKYNET data archives.

Specific Comments:

C2-2. Abstract: This is likely the shortest abstract I have ever seen, and hardly informative. I recommend that a few specifics be mentioned as there is really nothing of substance in this abstract.

A2-2. We extended the abstract summarizing the results of the paper.

C2-3. Page 3, lines 4-5: Here you discuss the nations that contribute to SKYNET and that a committee was formed for collaboration. Please add information on the availability of data to the scientific community from these regions/sub-networks and include specific data access websites and data policies.

A2-3. We added a new Table 2 to list geophysical parameter products, versions of Skyrad.pack, and availability for sub-networks.

C2-4. Page 3, lines 4-5: It is necessary (here or elsewhere in the text) to include information on the temporal frequency of the direct sun and sky radiance measurements for SKYNET sun-sky radiometers. Also important are the types of sky scans made by SKYNET instruments. Are they all almucantar and/or some additional solar principal plane scans? How often are the sky scans made, and over what solar zenith angle (SZA) range? This should be contrasted with the AERONET measurement protocols, almucantar and hybrid scans. Also over what SZA range are SKYNET almucantars identified as high quality retrievals. For AERONET the minimum SZA for Level 2 (high quality) almucantar retrievals is 50 degrees, corresponding to 100 degrees scattering angle range.

A2-4. We added more information about measurement protocols, theoretical basis, QC screening processes, and analysis algorithms with some comparison with those of AERONET to our knowledge in Section 5.

C2-5. Page 3, lines 4-5: It is surprising that the interference filters for the UV wavelengths would have such a wide band-pass of 10 nm. AERONET filters in the UV have 2 nm band-pass due to solar zenith angle dependent Rayleigh OD issues in measurement of AOD in the UV region.

A2-5. We are sorry that our original manuscript was inaccurate about the band-pass filter specifications. The POM 02 model uses 3 nm or less widths for 315, 340, and 380nm channels and 20nm for short IR channels. We added the numbers in Section 2.

C2-6. Page 4, lines 17-18: A short summary of the SKYNET cloud screening is warranted here in the text. It would be very useful to compare what aspects of the cloud screening are similar and which ones differ from the AERONET cloud screening (please refer to V3 cloud screening, discussed in Giles et al (2019)). For example, AERONET uses the angular steepness of the solar aureole as a cirrus filter, while I think that SKYNET does not.

A2-6. We added text to explain the cloud screening procedure and other QC protocols in Section 2 and 5 with text regarding the AERONET steepness test of the solar aureole referring to

Giles et al. (2019).

C2-7. Page 5, line 16: Regarding the improved Langley (IL) method, please provide some estimated ranges of accuracy for different optical depth magnitudes. Also provide accuracy estimates of Fo determination for temporal variation in AOD over the Langley measurement sequence, as it would be expected that this would also be a factor.

A2-7. We added a new Table 3 to list mean values of $n$ (number of observation points), AOT, and $\sigma_{a,IL}$ (uncertainty in $\ln F_0$) per 30 days (month) obtained from ILP operations carried out at Tokyo and Rome sites. For this explanation, we added a $\gamma$-value, sqrt$((\varepsilon_\tau/<\tau>)^2+(\varepsilon_\omega/<\omega>)^2)$, in new Eq. (10b). The $\gamma$-values for Tokyo and Rome are 7% and 15%, respectively, and close to 10% assumption for Eq. (10) in the original manuscript, but indication of 10% may cause a misleading idea to readers about the realistic accuracy, so we omitted the text regarding 10% assumption.

As for temporal change during ILP, the method allows such temporal change, as already stated in the original manuscript with an example of a temporal variation in AOT and SSA $\tau_1= 0.2$ to $\tau_2= 0.4$, and from $\omega_1= 0.85$ to $\omega_2= 0.95$ during ILP which is useful to increase the accuracy through Eqs. (7b) and (8a). It is also possible to have a change in the atmospheric conditions during a short time less than 5 min for one full angle scan for ILP to cause unexpected errors. Sub-networks, therefore, have their own screening protocols for ILP using stability of time sequence of variables to reject ill condition data for ILP. We added this point in the paragraph after Eq. (9).

C2-8. Page 6, line 13: Should be 'wasting' instead of 'waisting' here.

A2-8. Thanks. We corrected.

C2-9. Page 6, line 20: Since the AOD is much larger (and more variable) at 400 nm than 870 nm it seems that the error would be larger for the shorter wavelengths. Please explain why this would not be a factor or else include a statement that errors would be larger at shorter wavelengths unless the aerosol is coarse mode dominated (Angstrom Exponent of ←⋯zero).

A2-9. This is an important comment. It is true that ILP and AOT uncertainties increase with decreasing wavelength. The original intent was to state that the error relative to AOT is expected to be independent of wavelength by Eq. (10). But, this be misleading, so that we eliminate "regardless of wavelength" and instead we added new texts and new Tables 3 and 5 to give more observed numbers.

C2-10. Page 7-8 (last 2 sentences of P7 and first 2 sentences of P8): This data in Figure 4b

should be discussed in more detail in the next section of Sky radiance calibration (Section 4). Section 4, starting on Page 9: Please provide some explanation of the causes of the temporal variance of the SVA as shown in the time series in Figure 4b. This data is from high altitude sites with stable very low AOD and therefore would be expected to be a best case scenario. Also discuss what the variability in SVA from sky scans would look like at a sea level site with moderate and variable turbidity, say AOD=0.5 +-0.3 at 500 nm with alpha=1.5.

A2-10. The analysis of Fig. 4 was made by the Skyrad pack software for data screened by a condition of RMSD of SVAs is below 0.20, while the median value of the long-term data is much as 0.05. The observations were taken from a wide range of AOT with minimum (instantaneous) 0.01 to maximum 0.22 with the yearly averaged AOT as 0.0448±0.026 at 500 nm during 2008 to 2018 at the two sites. Due to limiting cloudy conditions in the afternoon hours, 35% of the disk scanning work are performed in between 8-9am. Since the disk scanning procedure takes around 20-25 minutes to complete the entire wavelengths, it is apparent that in some cases, some wavelengths may have been affected by thin (cirrus) clouds which carried by heavy wind (above 15 m/s) at both the sites.
We added this explanation in the related paragraph.

Uchiyama et al. (2018b) discussed that the SVA error by the disk scan can exceed 1% for large AOT condition like AOT550>0.5 and proposed the subtraction method using sky radiance calculated from the size distribution retrieved from the relative radiance. This subtraction method can reduce the error to 0.5% for AOT550< 2 for sky radiance measurement with the minimum scattering angle $\Theta$= 3°. So far, no sub-networks implement these methods in their operational analysis, but they reject large AOT cases from their disk scan data analysis by AOT value.
We added this explanation to the bottom paragraph of Section 4.

C2-11. Page 10, lines 13-15: It would be useful to provide the reader with an estimate of the magnitude of the error in sky radiance calibration as a result of this uncertainty in SVA from the disk scan. Or else state that the entire data base of SKYNET has been reprocessed with the correction method that was outlined by Uchiyama et al. (2018b).

A2-11. We added an error budget discussion for SSA retrieval by Pandithurai et al. (2008), Hashimoto et al. (2012), and Khartri et al. (2016) with new Table 6. According to these studies, the major error source of SSA retrievals by SKYNET is the underestimation of SVA by the disc scan method.

C2-12. Page 10, lines 20-24: Why are the retrievals within SKYNET made with two different

versions of the Skyrad.pak code (Version 4.2 or version 5)? What are the effects of these different codes on the inverted parameters? Why is there no standardized inversion code for the entire network? Or if I have mis-interpreted this then please clarify in the text what the consistent data processing is for the data from all sites in the SKYNET network.

A2-12. SKYET is a research network without a unified operation system for analyzing all the data using a unified analysis software. We added this statement at the bottom of the introduction section.

Due to historical reasons, Skyrad.pack version 4.2 and 5 are used in parallel. Also the two versions adopt different cost functions as explained by new Eq. (21), so that we still have a benefit from version 4.2.
We added the information to describe the version differences in the two versions in Section 5 and added new Table 6 to quantify the version difference for SSA retrievals.

C2-13. Page 11, line 1: Please state here whether this Cimel instrument in Beijing is part of the AERONET network with their data processing algorithms or a part of CARSNET network with somewhat different algorithms (especially when compared to V3 of AERONET).

A2-13. We added a text: In Che et al.(2008), the AOTs were compared between POM-02 skyradiometer and Cimel sunphotometer at the top of Institute of Atmospheric Physics(IAP) in Beijing which belongs to SKYNET and AERONET, respectively. The POM-02 data were processed by Skyrad pack 4.2. The sunphotometer data is AERONET V2 Level 2 data.

C2-14. Page 11, lines 2-3: Please explain here why the error in AOD is smaller in the shorter wavelengths even though the AOD and its absolute temporal variance is larger at the shorter wavelengths. This seems to be counter-intuitive, and I would need some significant evidence presented in the paper in order to be convincing.

A2-14. We rechecked the statement and found that the numbers are not RMSD, but the relative bias errors defined as <AOT(SKYNET)>-AOT(AERONET))/AOT(AERONET)). That is why the errors are very small as 1% and relatively independent of wavelength. We added new Table 5 to show their RMSD numbers that increase with decreasing wavelength. We added this explanation to the paragraph for new Table 5.

C2-15. Page 11, lines 10: This is a relatively large difference in Angstrom Exponent (0.5), but not if AOD is very low. Please state the AOD levels for these AE comparisons.

A2-15. Under low aerosol conditions, a small relative bias in the AOT determination at 500 and 865 nm can theoretically lead to large deviations in the calculated Ångström Exponents (AE). As an example, for AODs of about 0.05 and 0.02 at 500 and 865 nm, respectively, AOT differences of 0.01 and 0.005, respectively, can lead to AE differences up to ~1. We added this statement to the paragraph.

C2-16. Page 11, lines 13: It would be useful to also show or discuss comparison statistics of AOT for 380 nm (in addition to 870 nm in Figure 8) since all instrument types have larger uncertainties at this wavelength.

A2-16. The PFR used in the intercomparison campaigns did not have 380 nm channel available, therefore we cannot state the differences in channels below 400 nm. Instead, we compared, in the new Table 5, AOT retrievals from the POM-02 sky radiometer and the Cimel sunphotometer in the the KORUS-AQ campaign. They found RMSD from 0.007 to 0.15 at wavelengths longer than 500nm, but values larger than 0.03 are found in UV channels, as the reviewer pointed out.
We added this discussion in the related paragraphs for Table 5 and Fig. 8. We replaced Fig.8(b) from Roman result to a Davos result, because the new corona virus lock-down in Rome made it difficult to retrieve detailed data from the university where the data are stored.

C2-17. Page 11, lines 17-18: When discussing differences in AOD it is much more useful to provide the differences in AOD rather than percentages.

A2-17. We presented absolute differences in AOT in new Tables 3 and 5 and those of SSA in new Table 6 and reorganized the text in the related paragraphs.

C2-18. Page 11, lines 19: Provide some quantification here of the SSA overestimate found by Che et al. (2008).

A2-18. Che et al.(2008) found RMSD of 0.025, 0.018, 0.018, 0.018 at $\lambda$= 440, 670, 870, 1020nm, respectively. We compared the numbers to others in new Table 6 and added some discussion using Hashioto et al. (2012), Dubovik et al. (2000), and Khatri et al. (2016).

C2-19. Page 11, lines 23: There needs to be more discussion of the SKYNET cloud screening algorithms and comparison to co-located AERONET site data in order for users to understand the cloud contamination issue better.

A2-19. We added text to explain the cloud screening procedure and other QC protocols in

Section 2 and 5.

C2-20. Page 11, lines 23-25: It is puzzling as to why all the data are not processed with Version 5. This suggests that data at some sites that process the data with V4.2 will have more cloud contamination than other sites that use V5. Please provide some clarification on which versions of SKYRAD.pak software/algorithms are used to process the data for what sites.

A2-20. As answered in A2-12, SKYET is a research network without a unified operation system for analyzing all the data using a unified analysis software. We added this text at the bottom of the introduction section. By this historical reason, Skyrad.pack version 4.2 and 5 are used in parallel. Also the two versions adopt different cost functions as explained by new Eq. (21), so that we still have a benefit from version 4.2. We added text to describe the version differences in Section 5 and in new Table 6.

C2-21. Page 11, line 28: More details on the stricter cloud screening in V5 are needed here in the text, not just providing literature references.

A2-21. We eliminated the points (1)-(3), and instead we added text to explain the cloud screening and QC processes in ESR and CEReS in the second paragraph after Eq. (2c). And added discussion of the data screening protocols (C1)-(C3) to reject cloud contamination in the version 4.2 data analysis in Section 5.

C2-22. Page 12, lines 1-3: Your discussion of SSA differences (Fig 9) is misleading since the agreement was much poorer with standard SKYNET retrievals of SSA. The study of Mok et al. (2018) replaced the crude surface reflectance assumptions utilized by SKYNET with values derived from MODIS measurements (these were AERONET values). A discussion of the spectral surface reflectance used as input to the SKYNET retrievals is needed here.

A2-22. We added substantial text and a new Table 6 to compare reported SSA differences to show the SKYET SSAs are larger by more than 0.06 and can be reduced to around 0.03, if we introduce the various improvements including the spectral reflectance by AERONET. The spectral reflectance effect is 0.004 to 0.008 in the short wavelengths compared to the fixed albedo case in SKYNET. The Skyrad pack assumes simplifications regarding the extinction model and the inversion process as compared to those of AERONET, which may need to be implemented after more investigation. We added this statement in the related paragraphs regarding Tables 5 and 6, and conclusion section.

C2-23. Page 12-13, last 2 lines of P12 & first 4 line of P13: Please note whether this is a

SKYNET retrieval product for W (column water vapor) or just a possibility for future implementation. Also does SKYNET provide W retrieved from both methods shown in Fig 11 in the database for all sites?

A2-23. So far there are no sub-networks which produce an operational product of water vapor. We stated this point in Section 2 adding the new Table 2 to list what are the operational products. Figure 3 is also modified to show which are operational products. The method of Momoi et al. (2019) is just being applied to data in CEReS, but this point is too early to report in this paper. In this regard, we modified Eq. (26) (old Eq. (22)), which shows Momoi's new method, to the x-variable used in the classical methods.

C2-24. Page 13, line 11: It is very surprising that you use a 10 nm wide filter at 315 nm as stated on page 4 in this paper. This should cause significant SZA dependence in the signal and result in SZA dependence of the retrieved columnar ozone.

A2-24. Our original manuscript was inaccurate about the band-pass filter specifications. The POM-02 model uses 3 nm or less for 315, 340, and 380nm and 20nm in 1600nm and 2200nm channels. We added the numbers in Section 2.

C2-25. Page 13, lines 26-28: With only 1% of the variance explained in this scatterplot (Fig 13b), it seems that there are likely other issues involved in the retrieval of CER from both the sky radiometers and also the satellite measurements. The result could not be any worse than this and therefore it seems like this could just be mentioned in the text without the need of a plot.

A2-25. We agree with the reviewer's suggestion and dropped Fig. 13(b) for CER and added a text to state Khatri et al. (2019) also did not find a good correlation between retrieved cloud effective particle radius between SKYNET and AHI observations. We also revised Fig. 13 and Figs. 14(a) and (b) adding regression lines.

C2-26. Page 14, line 2: This statement is confusing. The accuracy shown in the paper is better than 10% if the AOT is high. For example, if the AOD at a site averages 0.5 for a month then the accuracy of the IL calibration does not result in 0.05 uncertainty.

A2-26. This is for the accuracy of $F_0$ for AOT condition in the ILP operation, in which AOTs are selected by a threshold smaller than the mean AOT at the site. For example, ESR rejects the data AOT500 > 0.4. So, the typical errors in retrieved $F_0$ is less than 1%. We added new Table 3 to list the reported uncertainties in $F_0$ and revised the text as above.

C2-27. Page 14, lines 2-3: This 'conclusion' is also misleading since the 'accuracy' of SSA is very hard to define as there is no gold standard for SSA measurement to use as a benchmark. Additionally. as discussed before this value of 0.015 (difference with AERONET and Pandora) is not based on standard SKYNET retrievals but on retrievals made with much more accurate inputs of surface reflectance (AERONET values were used). Also, Mok et al. (2018) applied additional improved quality checks to SKYNET retrievals, not the standard SKYNET product. The way it is currently written would give most readers a false sense of the consistency of the SKYNET retrievals of SSA versus AERONET.

A2-27. We revised the text that Mok et al. used spectrally varying AERONET ground albedo and stated that they also found that a prefixed ground albedo Ag at 0.1 by SKYNET increased RMSD by 0.004 to 0.008 in the short wavelengths.
In order to avoid readers' misunderstanding, we also stated in the abstract, introduction, and conclusion sections that some of these improvements are still in research phase and not involved in the operational system.

C2-28. Page 14, line 8: This is the first time the size of the financial budget for SKYNET has been mentioned in the entire paper. It is therefore a bit strange and confusing to include budget considerations in the Conclusions section.

A2-28. We eliminated the budget consideration from the text.

C2-29. Page 14, lines 10-13: It is odd to put these alternate calibration methods in bullet form in the Conclusions section. Please format into a typical sentence structure.

A2-29. We modified the statement in a running form.

**4. Other revisions/corrections**
The following revisions/corrections were applied other than comments by reviewers.
4-1. We added two authors, Akihiro Yamazaki and Sujung Go, because we needed their data and discussion for producing answers to some of reviewers' comments.
4-2. We added Ministry of Ocean and Fisheries, Korea in Acknowledgments.
4-3. Accuracy of precipitable water vapor was 0.2 cm instead of typo 2 cm in the original manuscript.
4-4. Table numbering changed because of insertion of new Table 3; accordingly Table 3 became Table 4. We also added new Tables 5 and 6. Equation numbering also changed due to insertion of new Eq. (21)-(24).

4-5. We replaced Fig.8(b) from Roman result to Davos result, because, in the new corona virus lock-down in Rome, we had a difficulty to retrieve detailed data from the Italian university where the data are stored.

End of document